# Pervasive genetic interactions modulate neurodevelopmental defects of the autism-associated 16p11.2 deletion in *Drosophila melanogaster*

Janani Iyer[1], Mayanglambam Dhruba Singh[1], Matthew Jensen[1,2], Payal Patel [1], Lucilla Pizzo[1], Emily Huber[1], Haley Koerselman[3], Alexis T. Weiner [1], Paola Lepanto[4], Komal Vadodaria[1], Alexis Kubina[1], Qingyu Wang [1,2], Abigail Talbert[1], Sneha Yennawar[1], Jose Badano [4], J. Robert Manak[3,5], Melissa M. Rolls[1], Arjun Krishnan[6,7] & Santhosh Girirajan [1,2,8]

As opposed to syndromic CNVs caused by single genes, extensive phenotypic heterogeneity in variably-expressive CNVs complicates disease gene discovery and functional evaluation. Here, we propose a complex interaction model for pathogenicity of the autism-associated 16p11.2 deletion, where CNV genes interact with each other in conserved pathways to modulate expression of the phenotype. Using multiple quantitative methods in *Drosophila* RNAi lines, we identify a range of neurodevelopmental phenotypes for knockdown of individual 16p11.2 homologs in different tissues. We test 565 pairwise knockdowns in the developing eye, and identify 24 interactions between pairs of 16p11.2 homologs and 46 interactions between 16p11.2 homologs and neurodevelopmental genes that suppress or enhance cell proliferation phenotypes compared to one-hit knockdowns. These interactions within cell proliferation pathways are also enriched in a human brain-specific network, providing translational relevance in humans. Our study indicates a role for pervasive genetic interactions within CNVs towards cellular and developmental phenotypes.

[1] Department of Biochemistry and Molecular Biology, The Pennsylvania State University, University Park, PA 16802, USA. [2] Bioinformatics and Genomics Program, The Huck Institutes of the Life Sciences, The Pennsylvania State University, University Park, PA 16802, USA. [3] Department of Biology, University of Iowa, Iowa City, IA 52242, USA. [4] Human Molecular Genetics Laboratory, Institut Pasteur de Montevideo, Montevideo CP11400, Uruguay. [5] Department of Pediatrics, University of Iowa, Iowa City, IA 52242, USA. [6] Department of Computational Mathematics, Science and Engineering, Michigan State University, East Lansing, MI 48824, USA. [7] Department of Biochemistry and Molecular Biology, Michigan State University, East Lansing, MI 48824, USA. [8] Department of Anthropology, The Pennsylvania State University, University Park, PA 16802, USA. These authors contributed equally: Janani Iyer, Mayanglambam Dhruba Singh, Matthew Jensen, Payal Patel. Correspondence and requests for materials should be addressed to S.G. (email: sxg47@psu.edu)

Rare recurrent copy-number variants (CNVs) with breakpoints typically mapping within segmental duplications are a significant cause of neurodevelopmental disorders, such as intellectual disability/developmental delay (ID/DD), autism, epilepsy, and schizophrenia[1]. Gene discovery within rare syndromic CNVs has traditionally involved mapping the disease-associated region using atypical CNVs, inversions, or translocations to identify a causative gene that explains the distinct phenotypes associated with the CNV, followed by detailed functional evaluation of that gene using animal models. Using this approach, the retinoic acid induced 1 gene (RAI1) was identified as the locus responsible for the core features of Smith-Magenis syndrome[2], and individual genes within chromosome 7q11.23 were connected to specific Williams-Beuren syndrome phenotypes, such as ELN for cardiovascular features[3]. Absence of atypical deletions for other CNVs required more direct functional evidence for implicating a candidate gene. For example, the role of TBX1 in aortic arch defects observed in individuals with 22q11.2 deletion/DiGeorge syndrome was identified through a functional screen for cardiac features in a series of mouse models carrying overlapping deletions of the human syntenic region[4]. All of these examples provide evidence that dosage alteration of one or more genes within the syndromic CNV interval contribute to the observed phenotypes.

Unlike rare CNVs associated with a consistent set of phenotypes, more recently described rare CNVs are associated with a range of neurodevelopmental features, and are also reported in unaffected or mildly affected individuals[1]. One such CNV is the 16p11.2 deletion, which encompasses 593 kbp and 25 unique genes. The deletion was originally identified in individuals with autism[5], and subsequently reported in children with ID/DD[6], epilepsy[7], and obesity[8]. Several themes have emerged from recent studies on dissecting the role of individual genes within the 16p11.2 deletion region towards neurodevelopmental phenotypes. First, extensive heterogeneity and incomplete penetrance of the associated phenotypes adds additional challenges to genetic mapping strategies that use atypical variants. Second, while this deletion is enriched within various neurodevelopmental disease cohorts, exome sequencing studies of hundreds of individuals have not identified any individual genes within this region as causative for these diseases on their own[9–12]. Third, functional studies using cellular[13], mouse[14–17], and zebrafish models[18] have implicated several different genes within 16p11.2 in neurodevelopmental phenotypes. These findings suggest that the observed phenotypes in 16p11.2 deletion are not caused by haploinsufficiency of a single causative gene, but rather are modulated by multiple dosage-sensitive genes in the region, potentially through combinatorial mechanisms within pathways related to neurodevelopment. This model is also consistent with a recent observation that pathogenic CNVs are more likely to contain clusters of functionally related genes than benign CNVs[19], suggesting intra-CNV genetic interactions as a potential cause for CNV pathogenicity. Therefore, an approach that combines a systematic functional evaluation of each gene within 16p11.2 and its genetic interactions is necessary to identify key neurodevelopmental pathways and molecular mechanisms of disease.

Evaluation of gene interactions in neurodevelopment requires a system that is sensitive to genetic perturbations but, at the same time, allows for performing interaction studies in the nervous system without compromising viability of the organism. Drosophila melanogaster provides such a model, as developmental processes, synaptic mechanisms, and neural structure and signaling are conserved between flies and vertebrates[20]. In fact, neurodevelopmental disorders[21] such as Angelman Syndrome, Rett Syndrome, Fragile X syndrome, and ID[22] have been modeled

in flies, while several studies have used Drosophila models to test for genetic interactions[23–25]. We use the power of Drosophila melanogaster as a genetic model to perform a series of quantitative and high-throughput assays to systematically characterize phenotypes, function, cellular mechanisms, and interactions of conserved homologs of human 16p11.2 genes. Our data suggest a complex interaction model for disease pathogenicity, where multiple 16p11.2 genes are sensitive to dosage imbalance and participate in complex interactions that both enhance and suppress the phenotypic effects of each other within cellular proliferation pathways, and in turn are modulated by other genes in the genetic background.

## Results

**Multiple 16p11.2 homologs are involved in neurodevelopment.** We identified 14 fly homologs from the 25 human 16p11.2 genes (Supplementary Table 1), and used 31 RNA interference (RNAi) lines and tissue-specific GAL4 drivers to knockdown the expression levels of individual homologs ubiquitously or in neuronal, eye, or wing tissues (Figs. 1 and 2a). RNAi is an effective strategy to model partial reduction of gene expression, which in principle recapitulates the effect of a heterozygous microdeletion, and for high-throughput screening of genes for tissue-specific phenotypes. We used multiple independent UAS-RNAi transgenes targeting the same gene to validate our results (Supplementary Fig. 1a, Supplementary Data 1), and used stringent quality control to eliminate lines that showed phenotypes due to off-target or positional effects (Supplementary Fig. 1a). Using quantitative PCR, we measured the reduction in gene expression for each line with neuronal knockdown (using Elav-GAL4 > Dicer2 at 25 °C), and on average achieved ~50% reduction in gene expression for the tested 16p11.2 homologs (Supplementary Data 2). As this study is focused on studying the functional role of human genes in a fly model, we represent the identified fly homologs in the format of HumanGene$^{FlyGene}$—for example, MAPK3$^{rl}$.

We performed a series of quantitative assays on 16p11.2 homologs for more than 20 phenotypes that have been classically used to measure conserved developmental function in flies[21], and identified lethality and a variety of morphological phenotypes due to ubiquitous and tissue-specific knockdown (Figs. 1 and 2a). For example, seven homologs were lethal at the larval or pupal stage with ubiquitous knockdown, indicating that these genes are essential for viability and development in Drosophila[26]. We next performed pan-neuronal knockdown experiments and tested for several nervous system phenotypes, such as climbing assays for motor impairment and spontaneous seizures. We performed negative geotaxis experiments to measure locomotor function and identified dramatic reductions in the climbing ability of MAPK3$^{rl}$ (Fig. 2b) and ALDOA$^{ald}$ (Supplementary Fig. 2a) knockdown flies throughout the testing period. Since about 24% of individuals with 16p11.2 deletion manifest seizures[27], we next used a recently developed spontaneous seizure assay that assesses unprovoked seizures in their native state, which better recapitulates human seizures in Drosophila[28]. We found that MAPK3$^{rl}$, PPP4C$^{pp4-19C}$, and KCTD13$^{CG10465}$ knockdown flies were more likely to show seizure phenotypes compared to controls (Fig. 2c, Supplementary Fig. 2b).

We further examined deeper cellular features, including neuromuscular junction, dendritic arborization, and axonal targeting, to understand the molecular basis of the observed neuronal features. Drosophila neuromuscular junction (NMJ) is a well-established model for studying synapse growth defects, and alterations in NMJ architecture have been documented in genes associated with autism[21]. We found significant differences in

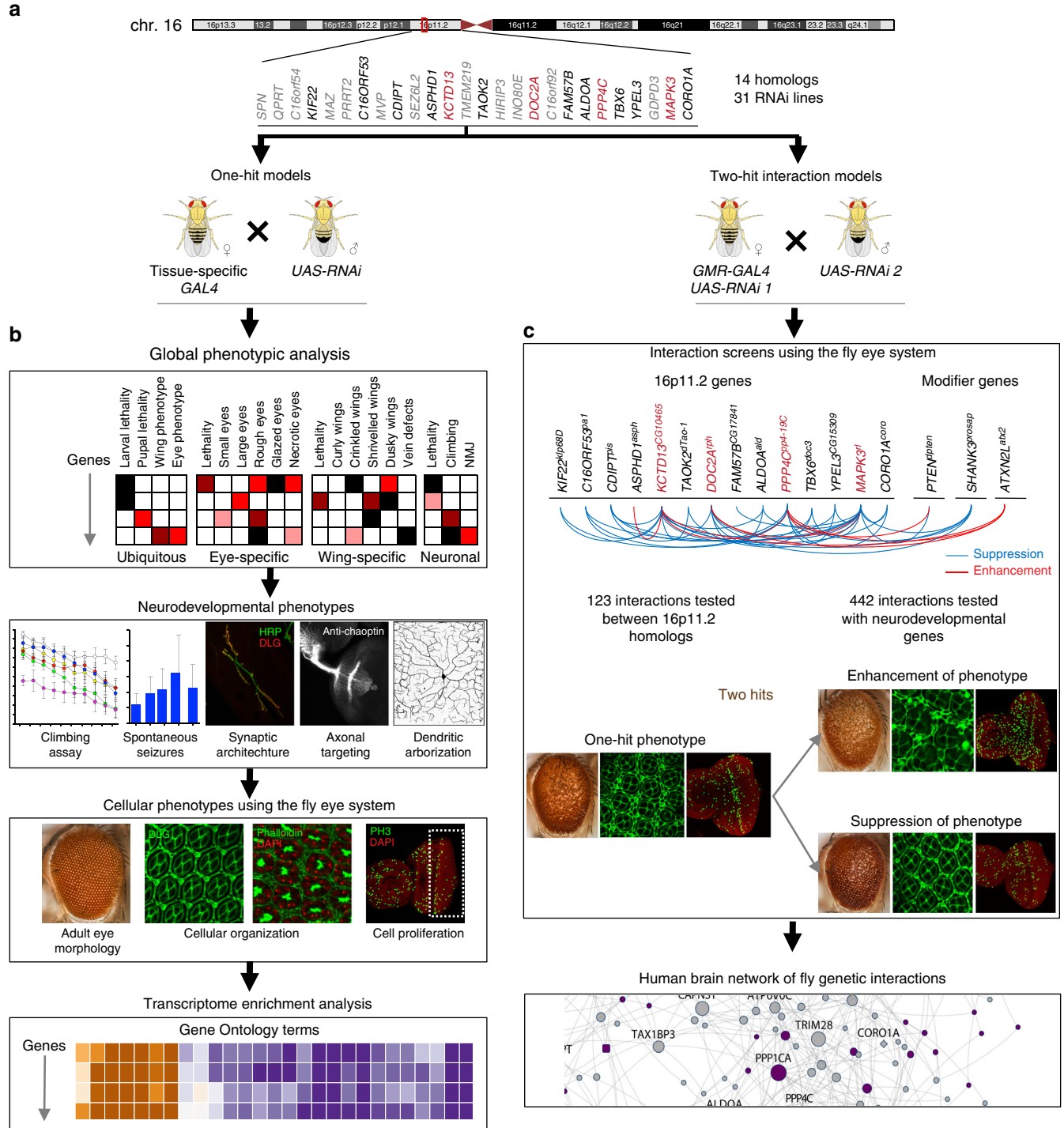

**Fig. 1** Strategy for identifying neurodevelopmental phenotypes in 16p11.2 fly homologs. **a** We identified 14 homologs of 16p11.2 deletion genes in *Drosophila melanogaster*, and **b** evaluated global, neurodevelopmental, and cellular phenotypes. We also performed transcriptome sequencing and assessed changes in expression of biologically significant genes. **c** Next, we identified modifiers of the one-hit eye phenotype for select homologs using two-hit interaction models. A subset of these interactions were further assessed for cellular phenotypes in the two-hit knockdown eyes. We incorporated all fly interactions into a human brain-specific genetic interaction network

NMJ structure at body wall muscles 6-7 with knockdown of *CDIPT^pis^* and *FAM57B^CG17841^* compared to controls, suggesting altered growth and development of the NMJ in these flies (Fig. 3a, Supplementary Fig. 2c and 2d). The architecture of dendritic arbors also plays an important role in neural circuit formation, and defects in dendrites are associated with neurodevelopmental disorders such as schizophrenia and autism[29].

To assay dendritic growth and structure, we examined large branched dendrites of the class IV ddaC sensory neuron in intact larvae after gene knockdown with the *ppk-GAL4* driver[30], and observed decreased complexity in dendritic arborization with knockdown of *KCTD13^CG10465^* and increased complexity with knockdown of *TAOK2^dTao-1^* (Fig. 3b, Supplementary Fig. 2e and 2f). Another hallmark of nervous

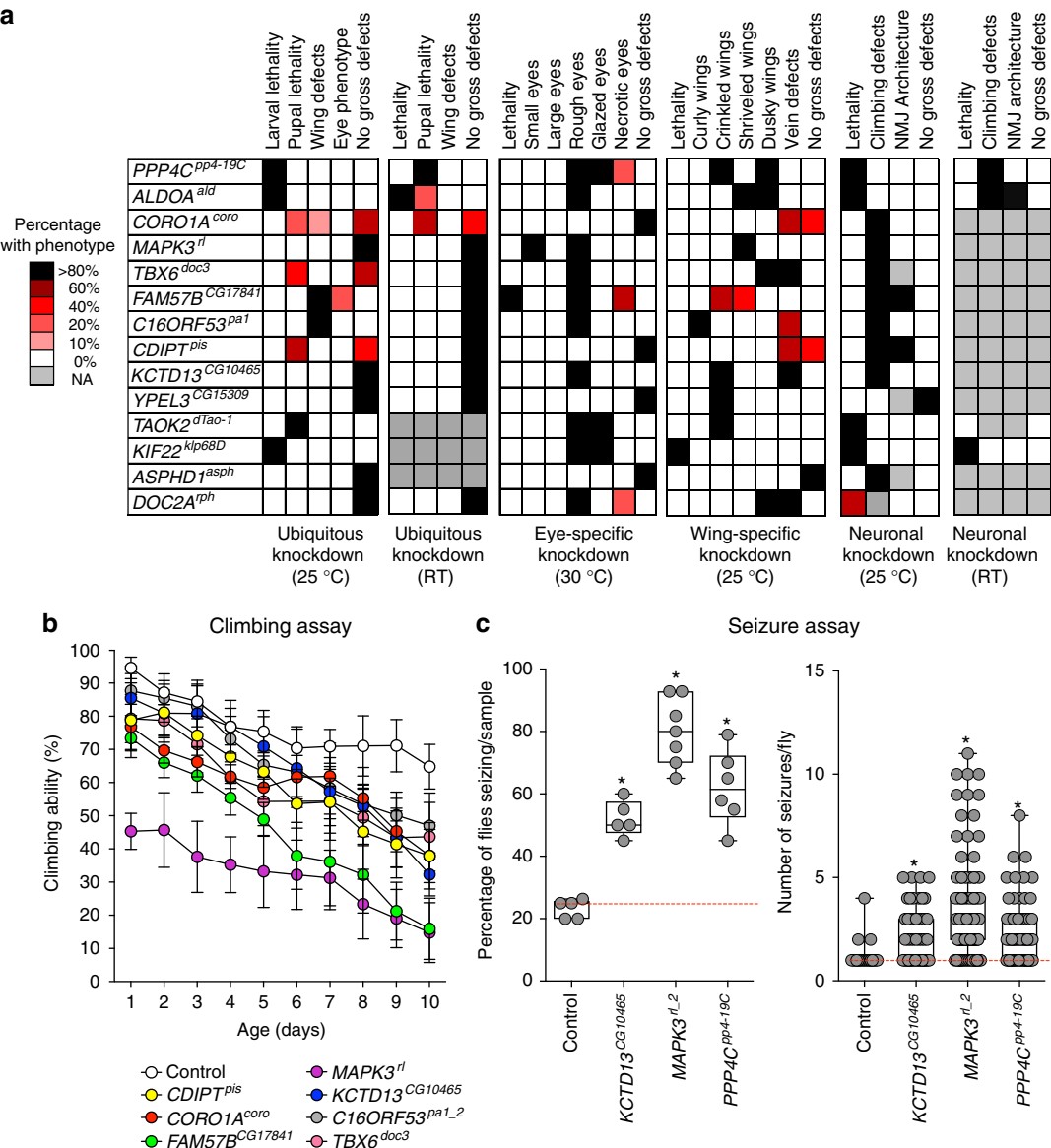

**Fig. 2** Neurodevelopmental defects in flies with knockdown of individual 16p11.2 homologs. **a** Percentage of 16p11.2 homologs with ubiquitous, eye-specific, wing-specific, and pan-neuronal knockdown at various temperatures that manifest specific phenotypes. **b** Assessment of 16p11.2 homologs for motor defects showed changes in climbing ability over ten days (two-way ANOVA, $p = 0.028$, df = 62, $F = 1.61$). Data represented here shows mean ± standard deviation of 10 independent groups of 10 flies for each line. **c** Assessment of knockdown of 16p11.2 homologs for frequency of spontaneous unprovoked seizure events ($n = 5–7$ replicate groups of 20 flies each) and average number of seizure events per fly ($n = 52–101$ individual flies, Mann–Whitney test, *$p < 0.05$). $PPP4C^{pp4-19C}$ knockdown was achieved using Elav-GAL4 and no Dicer2, and knockdown of the other two genes and the control used Elav-GAL4 > Dicer2. All boxplots indicate median (center line), 25th and 75th percentiles (bounds of box), and minimum and maximum (whiskers)

system development is the accuracy of synaptic connections, which is determined by the guidance of axons to their correct targets[31]. We explored axonal targeting by staining larval eye discs of flies using chaoptin antibody, and observed aberrant targeting in $KCTD13^{CG10465}$ and $MAPK3^{rl}$ flies (Fig. 3c). In summary, we found multiple developmental and neuronal defects for each of the homologs, indicating the pleiotropic effect of conserved 16p11.2 genes and their importance in neurodevelopment.

**Knockdown of 16p11.2 homologs leads to proliferation defects.** Decades of studies have shown that the Drosophila eye is an accessible and sensitized experimental system for quantitative studies of nervous system development and function, as genetic

defects that alter the development of a single cell type can lead to observable rough eye phenotypes[32] (Fig. 4). In fact, genetic interaction studies using the fly eye have led to the discovery of novel modifier genes in nervous system disorders, such as Rett syndrome and spinocerebellar ataxia type 3[21,33], as well as conserved developmental processes[34]. To quantify the degree of severity of the eye phenotypes, we developed and tested a computational method called Flynotyper that calculates a phenotypic score based on the disorderliness of ommatidial arrangement at high sensitivity and specificity[35]. We performed eye-specific knockdown of gene expression using the GMR-GAL4 driver with and without Dicer2 for all Drosophila 16p11.2 homologs, and compared the degree of phenotypic severity as measured by Flynotyper to controls with the same genetic background (Fig. 5a,c, Supplementary Fig. 3a–c). We found a strong

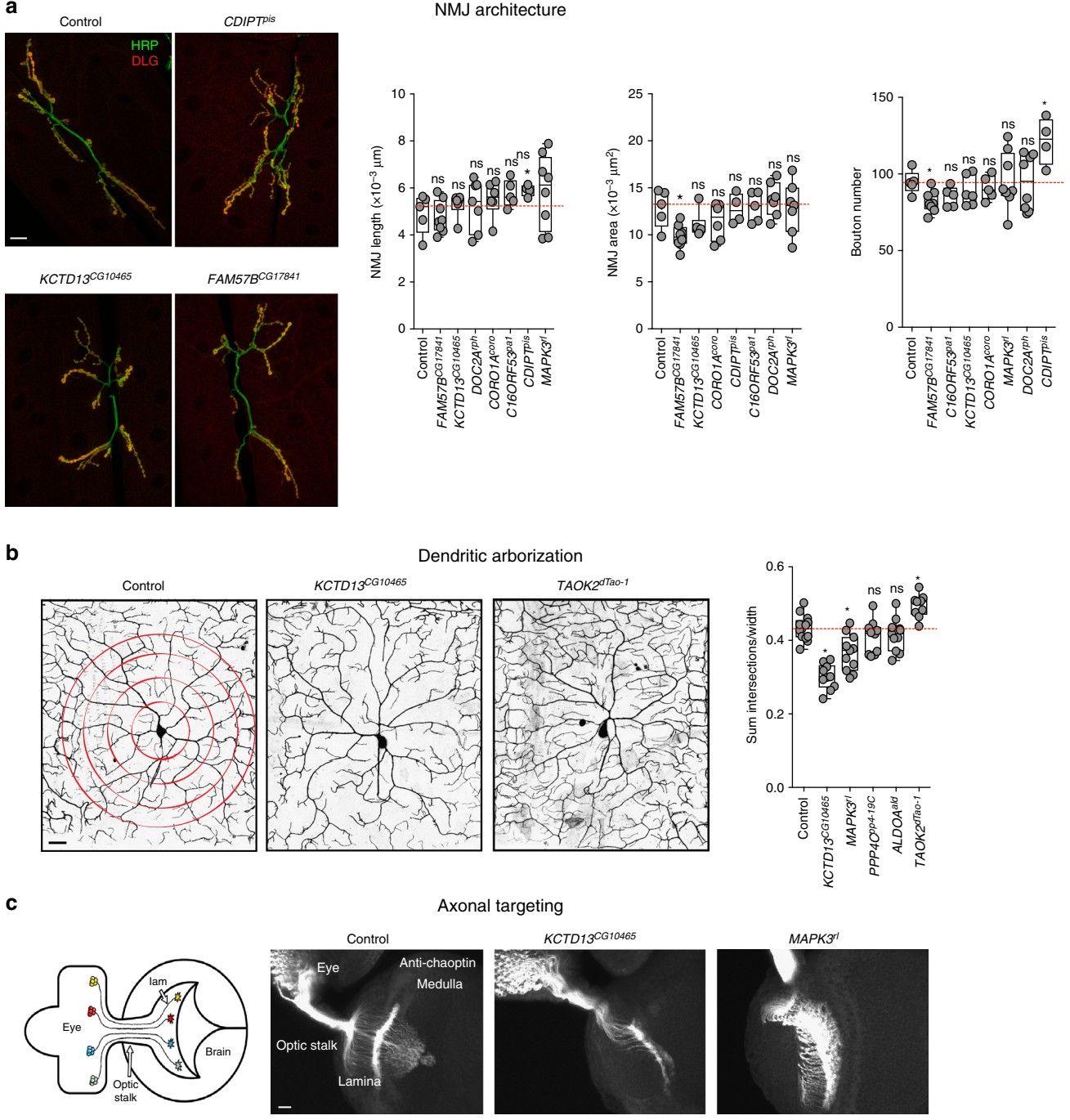

**Fig. 3** Neuronal phenotypes of flies with knockdown of individual 16p11.2 homologs. **a** Assessment of neuromuscular junction length, synaptic area, and bouton numbers for the tested 16p11.2 homologs ($n = 4$–8, *$p < 0.05$, Mann–Whitney test). Representative confocal fluorescent images (maximum projections of two or three optical sections) of the larval neuromuscular synapses are shown for three homologs (scale bar = 20 μm). **b** Assessment of dendritic arborization in larvae with knockdown of 16p11.2 homologs, including a box plot of the total number of intersections for all analyzed homologs, calculated by manual Sholl analysis and normalized to width measurement for each given hemisegment to control for slight size variation ($n = 9$–11, *$p < 0.05$, Mann–Whitney test). Representative confocal live images of class IV da neurons labeled with *mCD8-GFP* under the control of *ppk-GAL4* are shown for two 16p11.2 homologs and control (scale bar = 25 μm). **c** Assessment of axonal targeting with knockdown of 16p11.2 homologs. The schematic of the third-instar larval visual system was generated by Sam Kunes[68] and reprinted with permission from the publisher. Representative confocal images of larval eye discs stained with anti-chaoptin illustrate normal axonal targeting from the retina to the optic lobes of the brain in the control and defects with eye-specific knockdown of *KCTD13$^{CG10465}$* and *MAPK3$^{rl}$* (scale bar = 10 μm). All boxplots indicate median (center line), 25th and 75th percentiles (bounds of box), and minimum and maximum (whiskers)

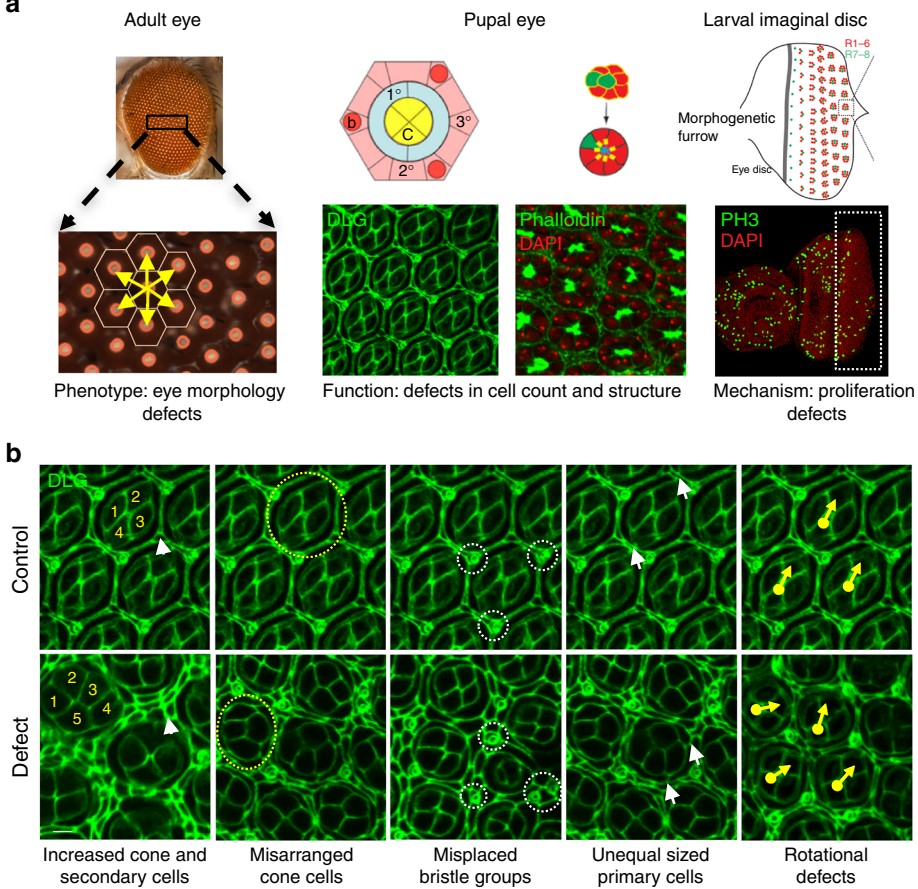

**Fig. 4** Screening strategy for neurodevelopmental defects in the developing fly eye. **a** Schematics and images of the wild-type adult, pupal, and larval eye show the cell organization and structure of the fly eye during development. The wild-type adult eye displays a symmetrical organization of ommatidia, and *Flynotyper* software detects the center of each ommatidium (orange circle) and calculates a phenotypic score based on the length and angle between the ommatidial centers. Illustrations of the wild-type pupal eye show the arrangement of cone cells (C), primary pigment cells (1°), and secondary pigment cells (2°) along the faces of the hexagon, and bristle cells (b) and tertiary pigment cells (3°) at alternating vertices, as well as the eight photoreceptor cells within an ommatidium. The larval imaginal disc schematic shows proliferating cells posterior to the morphogenetic furrow. Pupal eyes were stained with anti-Dlg and phalloidin to visualize ommatidial cells and photoreceptor cells, respectively, while the larval eye was stained with anti-pH3 to visualize proliferating cells. Diagrams of the pupal and larval eye were generated by Frank Pichaud[69] and Joan E. Hooper[70] and are reprinted with permission from the publishers. **b** Example images of pupal eyes stained with anti-Dlg illustrate the structure and organization in control and knockdown flies. Circles and arrows indicate differences in cell organization between control and knockdown pupal eyes (yellow circles: cone cell number and organization, white circles: bristle groups, white arrowheads: secondary cells, white arrows: primary cells, yellow arrows: rotation of ommatidia)

correlation (Pearson correlation, $r = 0.69$, $p = 2.88 \times 10^{-6}$) between the percentile ranks of all tested RNAi lines with *Dicer2* and without *Dicer2* (Supplementary Fig. 3d). As shown in Fig. 5c, we observed a range of severe but significant eye defects for nine homologs, which were comparable to that of genes associated with neurodevelopmental disorders such as *CHD8^{kis}*, *SHANK3-^{prosap}*, *SCN1A^{para}*, and *PTEN^{dpten}* (Supplementary Table 2, Supplementary Fig. 3d). For example, the severity of eye defects in *KCTD13^{CG10465}*, *DOC2A^{rph}*, and *PPP4C^{pp4-19C}* knockdown flies had phenotypic scores greater than the 85th percentile of the tested 39 fly homologs of human neurodevelopmental genes (Supplementary Table 2). These results suggested that knockdown of 16p11.2 homologs affect development of the fly eye to varying degrees of severity, which mirrors the global developmental and neurological defects observed with ubiquitous and pan-neuronal knockdown.

To investigate the cellular basis of the rough eye phenotype observed in individual gene knockdowns, we stained the pupal eye imaginal disc with Discs large (Dlg) antibody for ommatidial cells and Phalloidin for photoreceptor neurons, and screened for anomalies in different cells in the developing eye (Figs. 4 and 5a). A variety of cellular defects leading to altered structure of the hexagonal lattice were observed with knockdown of seven of the homologs, suggesting potential alterations in cellular proliferation (Fig. 5b, d, Supplementary Fig. 4). For example, *KCTD13^{CG10465}* knockdown flies showed a drastic increase in the number of cone and secondary pigment cells (Fig. 5b) and photoreceptor neurons (Fig. 5d), while *MAPK3^{rl}* knockdown showed a decreased number of photoreceptor neurons (Fig. 5d) and interommatidial cells, with a consequent loss of the hexagonal structure in the ommatidia (Fig. 5b). Similarly, *ALDOA^{ald}* knockdown flies had misplaced bristle cells (Fig. 5b) as well as an increase in secondary pigment cells and photoreceptor neurons (Fig. 5d), while *PPP4C^{pp4-19C}* knockdowns showed severe rotational defects and a complete loss of the ommatidial architecture (Fig. 5b). Overall, we found that knockdown of several 16p11.2 homologs contribute to defects in cell count and patterning of different cell types, including photoreceptor neurons and ommatidial cells, during development.

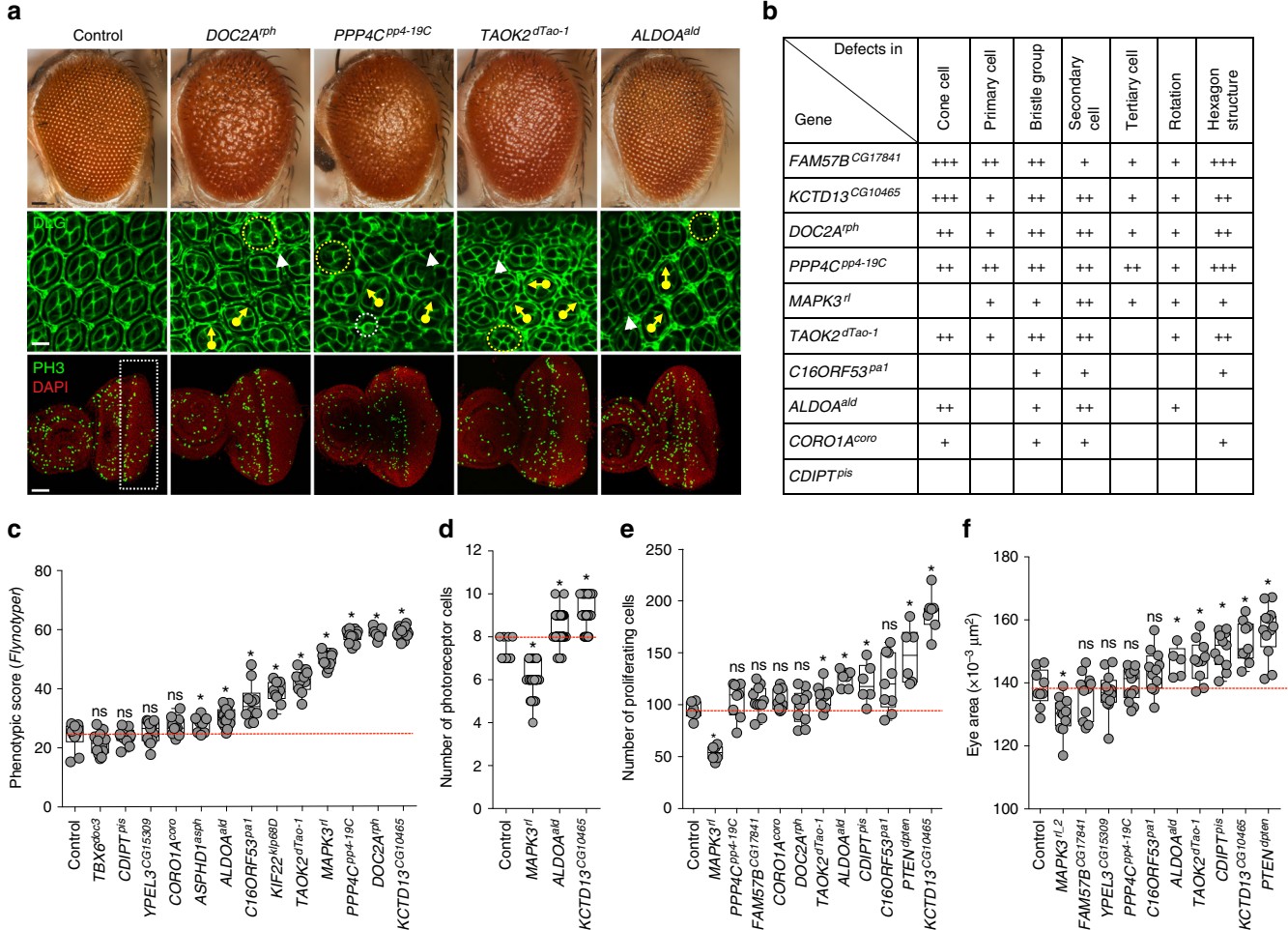

**Fig. 5** Cellular phenotypes in the fly eye due to knockdown of individual 16p11.2 homologs. **a** Representative brightfield adult eye images (scale bar = 50 μm) and confocal images of pupal eye (scale bar = 5 μm) and larval eye discs (scale bar = 30 μm), stained with anti-Dlg and anti-pH3 respectively, of select 16p11.2 homologs illustrate defects in cell proliferation caused by eye-specific knockdown of these homologs. **b** Table summarizing the cellular defects observed in the pupal eye of 16p11.2 homologs. "+" symbols indicate the severity of the observed cellular defects. **c** Box plot of *Flynotyper* scores for knockdown of 13 homologs of 16p11.2 genes with *GMR-GAL4 > Dicer2* (n = 7–19, *p < 0.05, Mann–Whitney test). *FAM57B*$^{CG17841}$ knockdown displayed pupal lethality with *Dicer2*, and therefore the effect of gene knockdown in further experiments was tested without *Dicer2*. **d** Box plot of photoreceptor cell count in the pupal eyes of 16p11.2 knockdown flies (n = 59–80, *p < 0.05, Mann–Whitney test). **e** Box plot of pH3-positive cell count in the larval eyes of 16p11.2 knockdown flies (n = 6–11, *p < 0.05, Mann–Whitney test). **f** Box plot of adult eye area in 16p11.2 one-hit knockdown models (n = 5–13, *p < 0.05, Mann–Whitney test). All boxplots indicate median (center line), 25th and 75th percentiles (bounds of box), and minimum and maximum (whiskers)

We further investigated cellular mechanisms associated with the observed developmental defects, as recent functional studies have implicated defects in neuronal proliferation as a cellular mechanism underlying autism disorders[36]. In fact, genome-wide CNV and exome sequencing studies of individuals with autism have uncovered pathogenic variants enriched for cell proliferation genes[37,38]. We used phospho-Histone H3 (pH3) antibody and bromodeoxyuridine (BrdU) staining to identify dividing cells, and counted the number of stained cells posterior to the morphogenetic furrow of the developing larval eye (Figs. 4a and 5a, Supplementary Fig. 4a and 4b). Several homologs showed a significant alteration in dividing cell counts (Fig. 5e). For example, we found an increase in mitotic cell count with knockdown of *KCTD13*$^{CG10465}$, *CDIPT*$^{pis}$, and *ALDOA*$^{ald}$, while *MAPK3*$^{rl}$ knockdown flies showed a significant reduction in proliferating cells (Fig. 5e). No changes in cell differentiation were observed using *Elav* antibody staining in *KCTD13*$^{CG10465}$ and *MAPK3*$^{rl}$ knockdowns, suggesting that these two genes are specifically involved in cell proliferation (Supplementary

Fig. 4b). Consistent with the proliferation phenotypes, we also observed an overall increase in the adult eye area in four RNAi knockdowns comparable to flies with knockdown of *PTEN*$^{dpten}$, a known cell proliferation gene[39], as well as a decrease in eye area for *MAPK3*$^{rl}$ flies (Fig. 5f). For *KCTD13*$^{CG10465}$ knockdown flies, we also found an increase in the size of ommatidia similar to that observed for *PTEN*$^{dpten}$ knockdown, indicating that cell growth defects in these flies may also occur with the observed increase in cell proliferation (Supplementary Fig. 4c). Overall, our analysis of individual gene knockdowns showed that reduced expression of individual 16p11.2 homologs cause defects in cell proliferation and organization.

**Interactions among 16p11.2 homologs modulate neurodevelopment.** Our phenotypic and functional studies of individual 16p11.2 homologs showed that many genes within the CNV region are involved in neurodevelopment, indicating that no single gene in the region is solely responsible for the

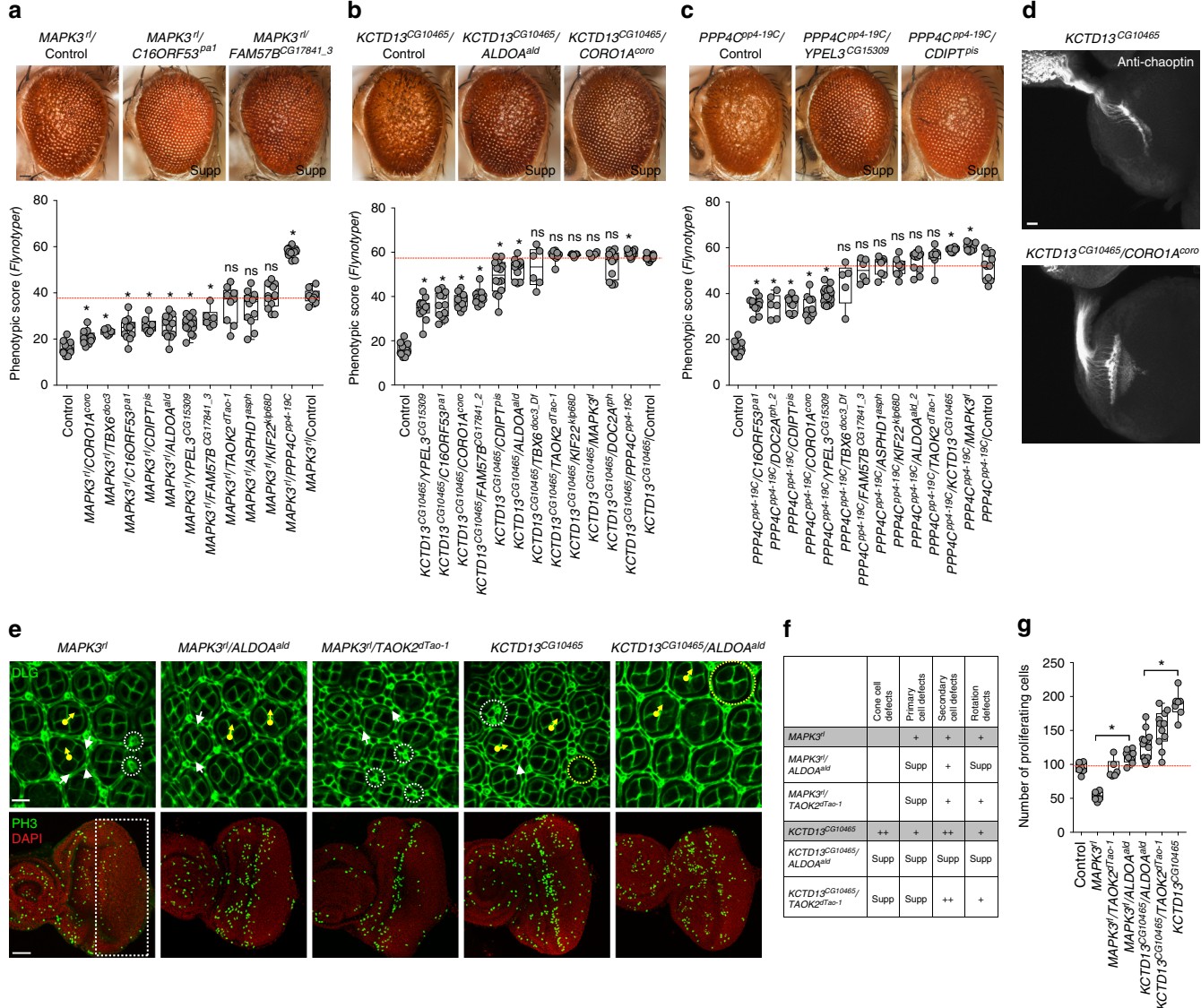

**Fig. 6** Phenotypic and functional effects of pairwise knockdown of 16p11.2 homologs. Representative brightfield adult eye images (scale bar = 50 μm) and box plots of *Flynotyper* scores of pairwise knockdown of **a** *MAPK3^rl* with other 16p11.2 homologs (*n* = 6–15, \**p* < 0.05, Mann–Whitney test), **b** *KCTD13^CG10465* with other 16p11.2 homologs (*n* = 4–14, \**p* < 0.05, Mann–Whitney test) and **c** *PPP4C^pp4-19C* with other 16p11.2 homologs (*n* = 5–17, \**p* < 0.05, Mann–Whitney test). **d** Assessment of axonal targeting in *KCTD13^CG10465/COROIA^coro* two-hit knockdown flies. Representative confocal images of larval eye discs stained with anti-chaoptin (scale bar = 10 μm) illustrate axonal targeting from the retina to the optic lobes of the brain in eye-specific knockdown of *KCTD13^CG10465*, and rescue of these defects with double knockdown of *KCTD13^CG10465* and *CORO1A^coro*. **e** Confocal images of pupal eye (scale bar = 5 μm) and larval eye discs (scale bar = 30 μm), stained with anti-Dlg and anti-pH3 respectively, for one-hit and two-hit knockdown of 16p11.2 homologs. **f** Table summarizing the cellular defects observed in the pupal eye of one-hit 16p11.2 flies compared to double knockdown of 16p11.2 homologs. "+" symbols indicate the severity of the observed cellular defects, while "Supp" indicates that the cellular defects were suppressed in the two-hit models. **g** Box plot of pH3-positive cell counts in the larval eye discs between one-hit and two-hit knockdowns of 16p11.2 homologs (*n* = 6-13, \**p* < 0.05, Mann–Whitney test). All boxplots indicate median (center line), 25th and 75th percentiles (bounds of box), and minimum and maximum (whiskers)

observed neuronal phenotypes of the deletion. Based on these observations, we hypothesized that interactions between genes within the 16p11.2 region may contribute to the observed phenotypes. To systematically test interactions between 16p11.2 homologs in the developing fly eye, we selected a subset of four homologs, including *PPP4C^pp4-19C*, *MAPK3^rl*, *KCTD13^CG10465*, and *DOC2A^rph*, and generated recombinant lines expressing their respective RNAi lines with the *GMR-GAL4* driver. We selected these genes as primary drivers of neurodevelopmental phenotypes based on severity of phenotypes with various tissue-specific knockdowns (Fig. 2a), published functional studies in mouse and zebrafish (Supplementary Table 1), and identifiable eye phenotypes amenable for large-scale modifier screens (Fig. 5c).

We generated 52 two-locus fly models (123 total lines) by reducing gene expression of each of the four homologs in combination with the 13 other 16p11.2 homologs. We used manual eye scoring and *Flynotyper* to compare these pairwise knockdown lines to respective control flies with single gene knockdowns. In this way, we identified 24 pairwise interactions

of 16p11.2 homologs, validated with multiple RNAi or deficiency lines that enhanced or suppressed the rough eye phenotypes observed with single-hit knockdown of the four tested genes (Fig. 6, Supplementary Fig. 5, Supplementary Data 3). Reduced expression of seven 16p11.2 homologs resulted in suppression of the rough eye phenotypes observed in $MAPK3^{rl}$ knockdown flies, including a full rescue of the $MAPK3^{rl}$ phenotype with

simultaneous knockdown of $CORO1A^{coro}$ and a partial rescue with knockdown of $C16ORF53^{pa1}$ and $FAM57B^{CG17841}$ (Fig. 6a). We also found that double knockdown of $MAPK3^{rl}$ and $PPP4C^{pp4-19C}$ led to an enhancement of the $MAPK3^{rl}$ rough eye phenotype (Fig. 6a). Similarly, reduced expression of six 16p11.2 homologs partially rescued the severe rough eye phenotype in $KCTD13^{CG10465}$ knockdown models, including $CORO1A^{coro}$ and

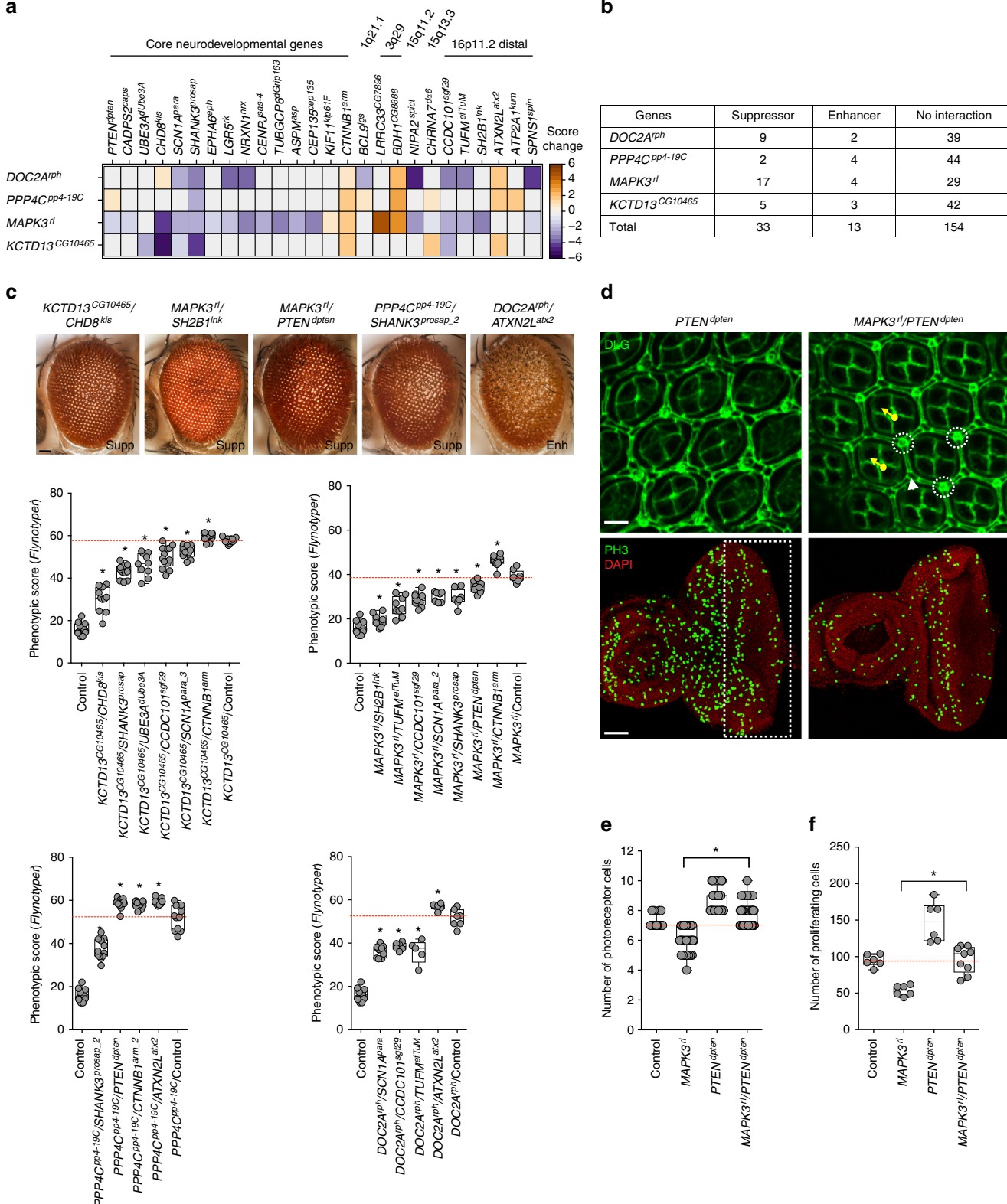

$ALDOA^{ald}$ (Fig. 6b). Further, the glossy eye phenotype observed in one-hit knockdown of $PPP4C^{pp4-19C}$ was suppressed by reduced expression of five homologs, including $YPEL3^{CG15309}$ and $CDIPT^{pis}$ (Fig. 6c). We also observed an enhancement of the rough eye phenotype with double knockdown of $PPP4C^{pp4-19C}$ and $KCTD13^{CG10465}$ at 30 °C, which we initially suspected to be due to the severity of $KCTD13^{CG10465}$ knockdown by itself (Supplementary Fig. 5d). To dissect this interaction, we performed reciprocal crosses of $KCTD13^{CG10465}$ RNAi lines with $PPP4C^{pp4-19C}$ at 25 °C, and confirmed the phenotypic enhancement observed at 30 °C (Supplementary Fig. 5d). Finally, the rough eye phenotypes in $DOC2A^{rph}$ knockdown models were suppressed by reduced expression of $CDIPT^{pis}$ and $ALDOA^{ald}$ (Supplementary Fig. 5e).

We tested a subset of 16p11.2 interaction pairs for alteration in their cellular phenotypes by staining the pupal and larval eye discs with anti-Dlg and anti-pH3, respectively (Fig. 6e, Supplementary Fig. 6a). Assessing the count, structure, and orientation of the cells in the developing eye discs confirmed several interactions documented in the adult eyes. For example, reduced expression of $ALDOA^{ald}$ in $MAPK3^{rl}$ models rescued the rotation errors and primary cell defects in the pupal eye (Fig. 6f, Supplementary Fig. 6c), as well as proliferation defects in the larval eye (Fig. 6g). Similarly, $ALDOA^{ald}$ knockdown suppressed the cone cell defects, secondary cell defects, and rotation errors observed in $KCTD13^{CG10465}$ knockdown pupae (Fig. 6f, Supplementary Fig. 6c), and showed a significant reduction in the number of proliferating cells compared to the $KCTD13^{CG10465}$ larval eye (Fig. 6g). Although reduced expression of $TAOK2^{dTao-1}$ did not rescue external eye defects in $MAPK3^{rl}$ and $KCTD13^{CG10465}$ knockdown flies, we observed partial rescue of cellular defects in the pupa (Fig. 6f, Supplementary Fig. 6c) and a significant rescue of proliferation defects in $MAPK3^{rl}$ and $KCTD13^{CG10465}$ larval eyes (Fig. 6g). To test if the two-hit interactions observed in the fly eye were also relevant in the nervous system, we evaluated the accuracy of retinal axon innervation into the lamina and medulla of the brain using anti-chaoptin in larvae with knockdown of both $KCTD13^{CG10465}$ and $CORO1A^{coro}$, and confirmed a complete rescue of the axonal targeting defects observed in single-hit $KCTD13^{CG10465}$ flies (Fig. 6d).

Some literature evidence exists for the functional interactions documented in our study. For example, MAPK3 and TAOK2 are both involved in synaptic assembly and signaling[40], and ALDOA was identified as a member of the MAP kinase (ERK1/2) interactome in differentiating epidermal and neuronal cells[41]. We also found that the tested 16p11.2 genes were connected to each other through intermediates at different degrees of separation within human gene interaction networks, potentially explaining the varying degrees of phenotypic modulation

observed in the two-hit fly models (Supplementary Fig. 6d and 6e). For example, the apoptosis regulatory gene[42] FKBP8 interacts with both KCTD13 and ALDOA in the brain, serving as an intermediate between these two genes. In fact, certain 16p11.2 human genes without fly homologs, including MAZ and HIRIP3, appeared as intermediate genes in our networks, further demonstrating the pervasive interactions between 16p11.2 genes. Overall, we found that pairwise knockdowns of 16p11.2 genes modulate cell proliferation defects observed in single-gene knockdowns during development. These defects can not only be enhanced but also rescued or suppressed by simultaneous knockdown of another 16p11.2 homolog, indicating that interactions between 16p11.2 homologs are epistatic in nature, where the phenotypic effects of two genes are greater or less than the sum of the effects of each individual gene[43]. These results point towards a new model for pathogenicity of the 16p11.2 deletion, where genes within the region are functionally related and interact with each other in conserved pathways to modulate the expression of the neurodevelopmental phenotype.

**16p11.2 homologs interact with known neurodevelopment genes**. To further explore the membership of 16p11.2 homologs within cellular and developmental pathways, we extended our two-hit interaction studies to include 18 homologs of known neurodevelopmental genes and 32 homologs of genes within five pathogenic CNV regions: 16p11.2 distal, 1q21.1, 15q13.3, 15q11.2, and 3q29. Using recombinant lines of $MAPK3^{rl}$, $KCTD13^{CG10465}$, $PPP4C^{pp4-19C}$, and $DOC2A^{rph}$, we tested a total of 200 pairwise gene interactions in 420 total lines (UAS-RNAi and deficiency lines) using manual scoring (Fig. 7a) and Flynotyper (Fig. 7c), and identified 46 interactions with 26 neurodevelopmental and CNV genes (Fig. 7b, Supplementary Fig. 7, Supplementary Data 4). Interestingly, 17 of these interactions of 16p11.2 homologs were with genes known to be involved in cell proliferation. For example, knockdown of the Wnt signaling pathway gene[44] $CHD8^{kis}$ resulted in a complete rescue of $MAPK3^{rl}$ phenotype (Supplementary Fig. 7a) as well as a strong suppression of the $KCTD13^{CG10465}$ rough eye phenotype (Fig. 7c). Similarly, reduced expression of beta catenin, $CTNNB1^{arm}$, significantly enhanced the phenotypes observed with $MAPK3^{rl}$, $KCTD13^{CG10465}$, and $PPP4C^{pp4-19C}$ one-hit knockdowns (Fig. 7c). Knockdown of $SHANK3^{prosap}$, a gene that codes for a post-synaptic scaffolding protein and is associated with autism[45], suppressed the rough-eye phenotype of $KCTD13^{CG10465}$, $MAPK3^{rl}$, and $PPP4C^{pp4-19C}$ flies (Fig. 7c). These data show that multiple 16p11.2 homologs interact through conserved neurodevelopmental genes that potentially act as intermediates, whose knockdown modulates the expression of the ultimate phenotype. Interestingly, six genes within the 16p11.2 distal CNV region interacted with 16p11.2 homologs. For example, reduced

**Fig. 7** Interactions of 16p11.2 homologs with neurodevelopmental genes. **a** Heatmap of change in phenotype measures (from manual scoring) in two-hit models of flies with knockdown of 16p11.2 homologs with core neurodevelopmental genes (left) or genes within CNV regions (right). Enhancers (orange) and suppressors (blue) for representative interactions of 16p11.2 homologs are shown. **b** Table summarizing the number of tested interactions of $DOC2A^{rph}$, $PPP4C^{pp4-19C}$, $MAPK3^{rl}$, and $KCTD13^{CG10465}$ with 50 neurodevelopmental and genes within other CNV regions. Of the 200 tested interactions measured by manual scoring or Flynotyper, 46 were identified as suppressors or enhancers of one-hit phenotype, and were validated in multiple RNAi or deficiency lines when available. **c** Representative brightfield adult eye images (scale bar = 50 μm) and box plots of Flynotyper scores for simultaneous knockdowns of $KCTD13^{CG10465}$, $MAPK3^{rl}$, $PPP4C^{pp4-19C}$, and $DOC2A^{rph}$ with neurodevelopmental genes ($n = 5$–13, $*p < 0.05$, Mann–Whitney test). **d** Representative confocal images of pupal eye (scale bar = 5 μm) and larval eye discs (scale bar = 30 μm) of the $MAPK3^{rl}/PTEN^{dpten}$ two-hit knockdown flies, stained with anti-Dlg and anti-pH3 respectively. **e** Box plot of photoreceptor cell count in the pupal eye of $MAPK3^{rl}$ and $PTEN^{dpten}$ one-hit and two-hit flies ($n = 58$–65, $*p = 3.62 \times 10^{-15}$ compared to one-hit knockdown of $MAPK3^{rl}$, Mann–Whitney test). **f** Box plot of pH3-positive cells in the larval eye between $MAPK3^{rl}$ and $PTEN^{dpten}$ one-hit and two-hit flies ($n = 9$, $*p = 0.00174$ compared to one-hit knockdown of $MAPK3^{rl}$, Mann–Whitney test). All boxplots indicate median (center line), 25th and 75th percentiles (bounds of box), and minimum and maximum (whiskers)

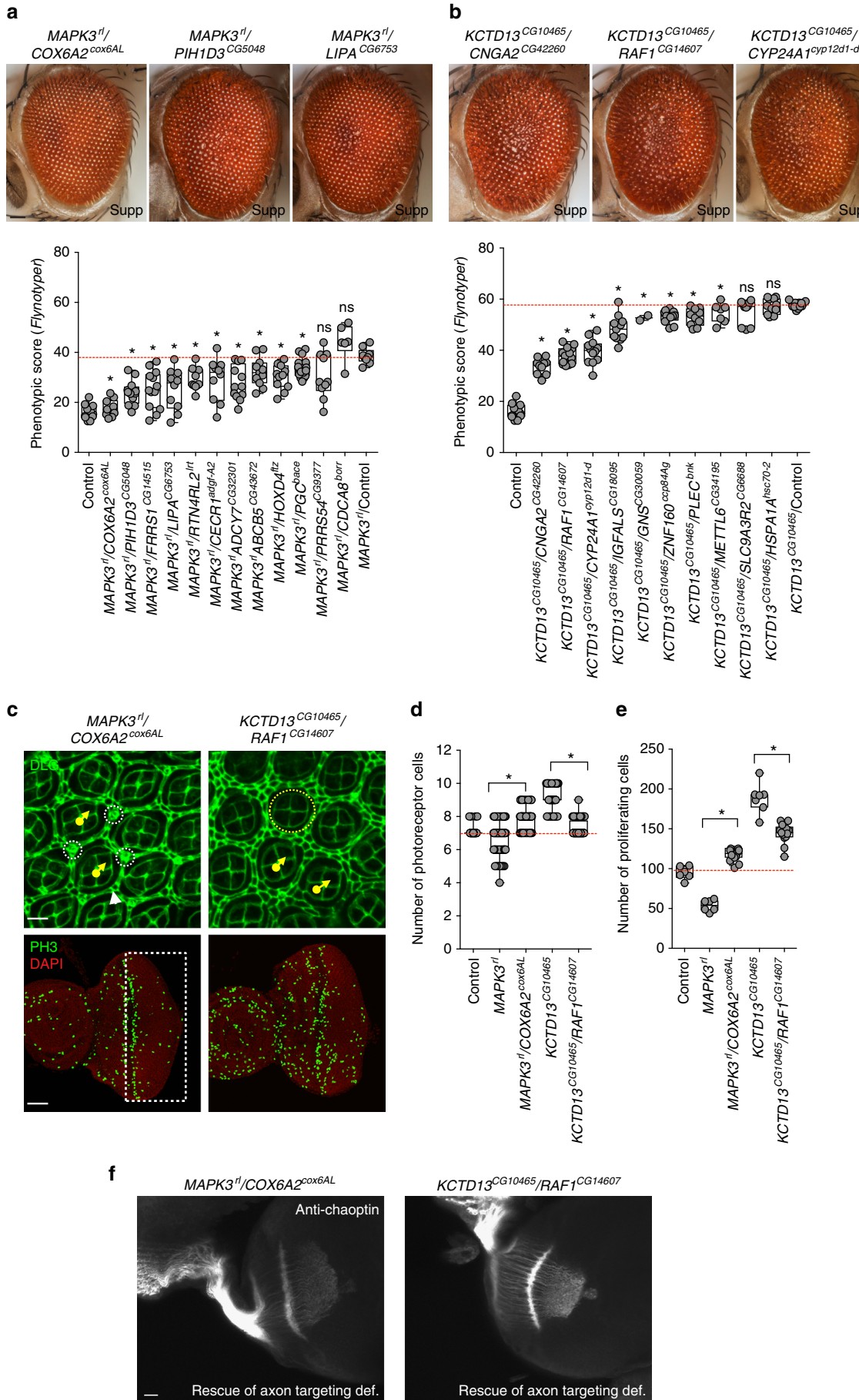

expression of $SH2B1^{lnk}$ fully rescued the rough-eye phenotype observed with knockdown of $MAPK3^{rl}$ (Fig. 7c), while $ATXN2$-$L^{atx2}$ knockdown led to a more severe phenotype in $PPP4C^{pp4-19C}$, $KCTD13^{CG10465}$, and $DOC2A^{rph}$ knockdown flies (Fig. 7c, Supplementary Fig. 7b). These results suggest overlapping functional roles in neurodevelopment for genes within the proximal and distal 16p11.2 regions apart from chromosomal contacts between the two syntenic segments in humans[46]. We further assessed the cellular mechanisms responsible for suppression of the $MAPK3^{rl}$ rough eye phenotype by simultaneous knockdown of the autism-associated tumor suppressor[47] $PTEN^{dpten}$ (Fig. 7c, d, Supplementary Fig. 7e). We observed a complete rescue of defects in cellular organization (Supplementary Fig. 7f), photoreceptor cell counts (Fig. 7e) and cell proliferation (Fig. 7f) observed with $MAPK3^{rl}$ single-hit knockdown.

**16p11.2 gene networks show enrichment for cell proliferation.** To examine functional processes associated with 16p11.2 homologs, we selected six 16p11.2 homologs, $MAPK3^{rl}$, $KCTD13^{CG10465}$, $DOC2A^{rph}$, $CORO1A^{coro}$, $C16ORF53^{pa1}$, and $CDIPT^{pis}$, based on high phenotypic severity, and performed RNA sequencing on fly heads for each knockdown to identify differentially-expressed genes (Supplementary Data 5). We first conducted parametric gene-set enrichment analysis[48] to identify Gene Ontology terms enriched for human homologs of up- or down-regulated genes in each fly model relative to the control (Supplementary Fig. 8a, Supplementary Data 5). Several terms related to neurodevelopment, including cell proliferation, cell cycle process, neurogenesis, neuron differentiation, neuron projection development, and cell–cell signaling, were significantly enriched in the knockdown models ($p < 0.01$, corrected by Benjamini-Hochberg method, Parametric Analysis of Geneset Enrichment test) (Supplementary Fig. 8a). Based on the cell proliferation phenotypes observed in the fly eye, we further constructed a network of differentially-expressed genes annotated for cell cycle and proliferation in humans (Supplementary Fig. 8b). Interestingly, we found a significantly high degree of overlap among differentially-expressed cell proliferation genes between the knockdown models (empirical $p < 0.001$), with 38.5% (65/169) of these genes differentially expressed in two or more models and 16.6% (28/169) differentially expressed in at least three knockdown models. These results provide additional evidence that the 16p11.2 homologs function in well-connected cell proliferation pathways in both *Drosophila* and humans.

We next selected 22 genes that were among the most up-regulated genes in $KCTD13^{CG10465}$ and $MAPK3^{rl}$ models for two-hit interaction experiments, and identified 18 genes whose knockdown suppressed the rough eye phenotypes in the $MAPK3^{rl}$ (Fig. 8a) and $KCTD13^{CG10465}$ flies (Fig. 8b). For example, knockdown of $COX6A2^{cox6AL}$ fully rescued the $MAPK3^{rl}$ rough eye phenotype (Fig. 8a), and knockdown of $CNGA2^{CG42260}$, $CYP24A1^{cyp12d1-d}$ and $RAF1^{CG14607}$

suppressed the $KCTD13^{CG10465}$ phenotype (Fig. 8b). We further examined the cellular phenotypes of $MAPK3^{rl}/COX6A2^{cox6AL}$ and $KCTD13^{CG10465}/RAF1^{CG14607}$ knockdowns by staining the larval and pupal eye discs with anti-Dlg, phalloidin, anti-pH3, and anti-chaoptin (Fig. 8c, f, Supplementary Fig. 7e). Defects in cone cells and primary and secondary pigment cells (Supplementary Fig. 7f), as well as photoreceptor cell counts (Fig. 8d) and proliferating cell counts (Fig. 8e), were all corrected in both two-locus models. Additionally, $RAF1^{CG14607}$ and $COX6A2^{cox6AL}$ knockdowns rescued the aberrant axonal targeting observed in $KCTD13^{CG10465}$ and $MAPK3^{rl}$ flies, respectively (Fig. 8f). While interactions between $RAF1$ and $KCTD13$ have not been previously reported, $COX6A2$ was shown to interact with $MAPK3$ within a human-specific gene interaction network[49]. $COX6A2$ was also differentially expressed in a 16p11.2 deletion mouse model[13], providing further evidence for this interaction. Of note, although the transcriptome analysis was performed on fly brains using a neuron-specific driver, we were able to validate those interactions in the fly eye, supporting the utility and veracity of using the *Drosophila* eye to study nervous system interactions.

To assess the relevance of the identified functional interactions in our fly experiments to neurodevelopment in humans, we explored the functional context of the human 16p11.2 genes and their involvement in cell proliferation, specifically in relation to brain biology, using a Bayesian network of known and predicted genetic interactions in the brain. We first mapped 14 homologs of 16p11.2 genes and 35 interacting genes identified from fly experiments (26 key neurodevelopmental genes and nine differentially-expressed genes from transcriptome studies) onto a human brain-specific gene interaction network[50,51], and then identified additional genes in the network that connected these genes to each other. Overall, we found 982 interactions present in the human brain, with 39 out of the 49 tested genes connected through 428 novel genes within the network (Fig. 9, Supplementary Data 6). A significant enrichment for cell cycle and cell proliferation function was identified among these novel connector genes (96/428, one-tailed Fisher's exact test, $p = 3.14 \times 10^{-12}$). However, we also found the same enrichment among connector genes for random sets of fly genes exhibiting neurological and behavioral phenotypes, indicating that this result is likely a general characteristic for genes involved in neurodevelopment. Additionally, our connector genes were enriched for genes related to neurodevelopment, including FMRP binding genes ($p = 3.34 \times 10^{-14}$, one-tailed Student's $t$-test) and genes involved in post-synaptic density ($p = 3.31 \times 10^{-32}$, one-tailed Student's $t$-test)[9], as well as genes differentially expressed in the fly knockdown models ($p = 0.0215$, one-tailed Student's $t$-test). These results suggest a strong concordance between data obtained from fly two-locus experiments with putative interactions identified in the human nervous system, and provide a novel set of candidates that could be potential therapeutic targets for the deletion phenotypes.

**Fig. 8** Interactions of 16p11.2 homologs with differentially expressed genes. Representative brightfield adult eye images (scale bar = 50 μm) and box plots of *Flynotyper* scores for **a** pairwise knockdown of $MAPK3^{rl}$ and up-regulated genes identified from transcriptome data ($n = 6$–13, *$p < 0.05$, Mann–Whitney test), and **b** pairwise knockdown of $KCTD13^{CG10465}$ and up-regulated genes identified from transcriptome data ($n = 2$–14, *$p < 0.05$, Mann–Whitney test). **c** Confocal images of pupal eye (scale bar = 5 μm) and larval eye discs (scale bar = 30 μm) stained with anti-Dlg and anti-pH3, respectively, for $MAPK3^{rl}/$ $COX6A2^{cox6AL}$ and $KCTD13^{CG10465}/RAF1^{CG14607}$ two-hit knockdown flies. **d** Box plot of photoreceptor cell counts in $MAPK3^{rl}/COX6A2^{cox6AL}$ and $KCTD13^{CG10465}/RAF1^{CG14607}$ two-hit knockdown flies ($n = 62$–68, *$p < 0.05$, Mann–Whitney test). **e** Box plot of the number of pH3-positive cells in $MAPK3^{rl}/COX6A2^{cox6AL}$ and $KCTD13^{CG10465}/RAF1^{CG14607}$ two-hit knockdown flies ($n = 12$–13, *$p < 0.05$, Mann–Whitney test). **f** Assessment of axonal targeting in $MAPK3^{rl}/COX6A2^{cox6AL}$ and $KCTD13^{CG10465}/RAF1^{CG14607}$ two-hit knockdowns. Representative confocal images of larval eye discs stained with anti-chaoptin (scale bar = 10 μm) illustrate rescue of axonal targeting defects in the two-locus models (compared to one-hits shown in Fig. 3c). All boxplots indicate median (center line), 25th and 75th percentiles (bounds of box), and minimum and maximum (whiskers)

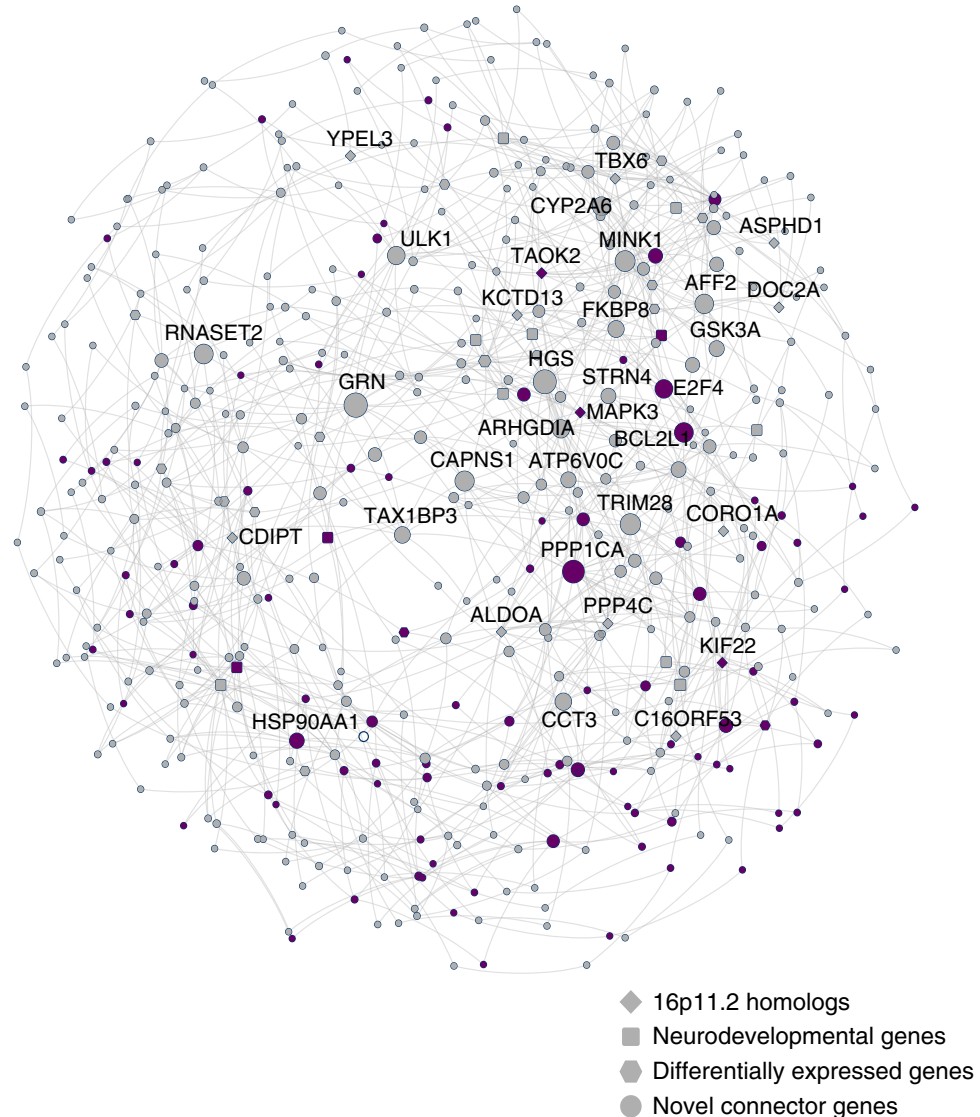

**Fig. 9** Human brain-specific network of 16p11.2 gene interactions. This network displays human brain-specific genetic interactions of all tested 16p11.2 genes and modifier genes, as well as neighboring connector genes. Network nodes with thick borders represent tested genes, with node shape representing gene category. The size of the nodes is proportional to how many connections they have in the network, and the thickness of the edges is proportional to the number of critical paths in the network using that edge. Purple nodes are genes annotated with cell proliferation or cell cycle GO terms

## Discussion

We used the sensitive genetic system of *Drosophila* to identify conserved functions and interactions of 16p11.2 homologs. While previous functional studies of the 16p11.2 region have either focused on the phenotypic effects of the whole deletion or the additive effects of individual genes, our work provides functional evidence that uncovers a complex model of genetic interactions in this region. The composite system of the fly eye allowed us to assay multiple genes and hundreds of interactions using high-throughput and quantitative assays for neurodevelopment without compromising the organism's viability. In fact, we were able to validate nervous system-specific interactions identified through transcriptome studies using both the fly eye and a human brain-specific network, which provides strong support for the use of genetic screens in the *Drosophila* eye for studying mechanisms of disease in the nervous system. Additionally, we identified multiple interactions of conserved 16p11.2 homologs that are consistent with published biochemical studies as well as functional gene networks constructed from human co-expression and protein–protein interaction datasets. Screening for interactions

with neurodevelopmental genes and differentially-expressed genes in the transcriptome could be particularly useful in identifying potential therapeutic targets for 16p11.2 deletion phenotypes. For example, rapamycin has been shown to rescue cellular and behavioral phenotypes in mouse knockouts of *PTEN*[52], while sorafenib inhibits growth and promotes apoptosis in cancer cells with *RAF1* mutations[53]. Therefore, therapeutic targets for the identified suppressors of multiple 16p11.2 homologs both within and outside the region, such as *ALDOA*, *CORO1A*, *CHD8*, *PTEN*, and *RAF1*, could be tested in full deletion models for a reduction in severity of neurodevelopmental phenotypes. This approach would be especially well suited for 16p11.2 deletion, where genes participating in a shared pathway can be targeted by a single treatment (instead of multiple targets for individual CNV genes). Although we observed interactions in the subset of 16p11.2 genes with homologs in *Drosophila*, it is likely that genes without fly orthologs also contribute to complex genetic interactions in the region. In fact, two 16p11.2 genes not tested in flies, *MAZ* and *HIRIP3*, were found in human gene interaction networks based on the tested homologs. Interactions between *MAZ* and other

16p11.2 genes are especially noteworthy, as a recent study reported cell proliferation defects in human embryonic kidney cells upon siRNA knockdown of *MAZ*[17]. Overall, our findings provide evidence for specific interactions in the 16p11.2 region that can be integrated with data from more sophisticated neurobiological systems, such as human stem cells, mouse, and zebrafish, to fully explain the complex interactions responsible for the neurodevelopmental phenotypes observed in 16p11.2 deletion carriers.

Multiple lines of evidence including two-hit screening in the fly eye, transcriptome data from fly heads, and human brain-specific genetic interaction network suggest that interactions among 16p11.2 genes are mediated through cellular proliferation and cell cycle pathways, which are well-conserved between flies and humans[54]. In fact, several 16p11.2 genes have already been implicated in these cellular pathways. For example, *MAPK3* is a key member of the MAPK/ERK signaling pathway, which is partially regulated by *TAOK2* and *ALDOA*[40,41], while *KCTD13* encodes polymerase delta-interacting protein 1 (PDIP1), which interacts with the proliferating cell nuclear antigen and therefore could have a role in the regulation of cell cycle during neurogenesis[55]. Our results are also consistent with aberrant changes in proliferation during early cortical neurogenesis observed in a 16p11.2 deletion mouse model[14]. While a recent study by

**a**

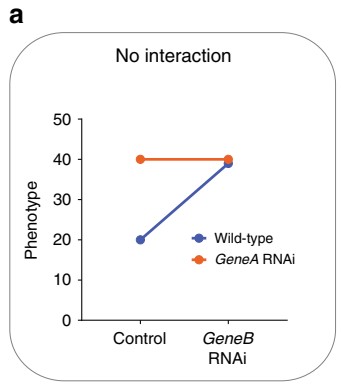
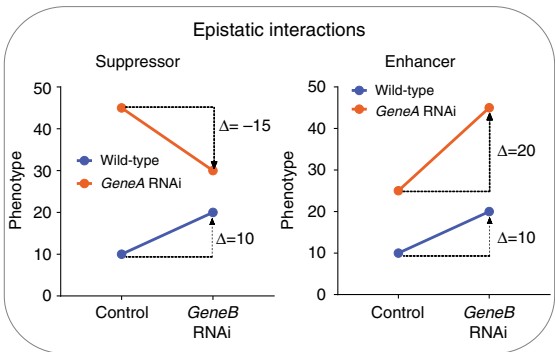
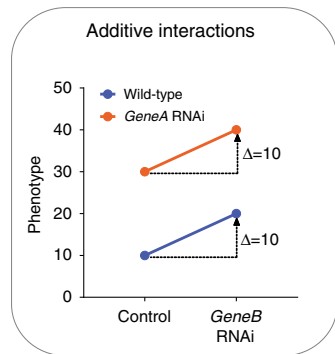

**b**

| | Total interactions | | Epistatic interactions | | | Additive interactions | |
|---|---|---|---|---|---|---|---|
| | Tested | Validated | 16p11.2 genes | Neurodevelopmental genes | RNA-Seq targets | 16p11.2 genes | Neurodevelopmental genes |
| *KCTD13*^CG10465 | 45 | 23 | 6: **C16ORF53**^pa1, CDIPT^pis, FAM57B^CG17841, **CORO1A**^coro†, ALDOA^ald·†, YPEL3^CG15309 | 5: CCDC101^sgf29, UBE3A^dUbe3A, CHD8^kis, SCN1A^para, SHANK3^prosap | 8: CNGA2^CG42260, **RAF1**^CG14607·†, CYP24A1^cyp12d1-d, IGFALS^CG18095, GNS^CG30059, ZNF160^ccp84Ag, PLEC^bnk, METTL6^CG34195 | 1: PPP4C^pp4-19C | 3: ATXN2L^atx2, **CHRNA7**^gfa6, **CTNNB1**^arm |
| *MAPK3*^rl | 47 | 39 | 7: **C16ORF53**^pa1, CDIPT^pis, FAM57B^CG17841, **CORO1A**^coro, ALDOA^ald·, YPEL3^CG15309C, TBX6^doc3 | 18: CCDC101^sgf29, TUFM^efTuM, **SH2B1**^lnk, SPNS1^spin, **PTEN**^dpten·, CHD8^kis, SCN1A^para, SHANK3^prosap, EPHA6^eph, **LGR5**^rk, NRXN1^nrx, **CEP135**^cep135, **CENPJ**^sas-4, **TUBGCP6**^grip128, **ASPM**^asp, NIPA2^spict, **CHRNA7**^gfa, ~~CTNNB1~~^arm | 10: COX6A2^cox6AL·†, PIH1D3^CG5048, FRRS1^CG14515, **LIPA**^CG8753, RTN4RL2^nrx, CECR1^adgf-A2, ADCY7^CG32301, ABCB5^CG43672, HOXD4^ftz, PGC^bace | 1: PPP4C^pp4-19C | 3: **KIF11**^klp61F, LRRC33^CG7895, BDH1^CG8888 |
| *PPP4C*^pp4-19C | 37 | 13 | 5: **C16ORF53**^pa1, CDIPT^pis, **CORO1A**^coro, DOC2A^rph, YPEL3^CG15309 | 2: SHANK3^prosap, **CHRNA7**^gfa | -- | 2: KCTD13^CG10465, **MAPK3**^rl | 4: ATXN2L^atx2, ATP2A1^kum, **PTEN**^dpten, **CTNNB1**^arm |
| *DOC2A*^rph | 37 | 13 | 2: CDIPT^pis, ALDOA^ald | 9: CCDC101^sgf29, TUFM^efTuM, SPNS1^spin, SCN1A^para, SHANK3^prosap, **LGR5**^rk, NRXN1^nrx, NIPA2^spict, BCL9^lgs | -- | 0 | 2: ATXN2L^atx2, **CTNNB1**^arm |

**c**

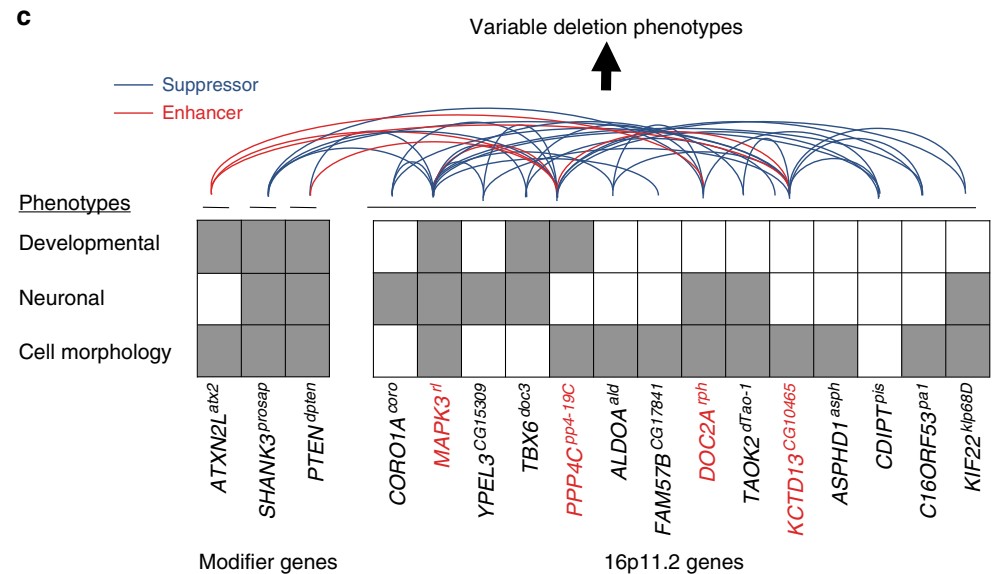

Deshpande and colleagues did not find defects in cell proliferation in their neural progenitor cells[56], the discrepancies could be attributed to testing individual 16p11.2 genes versus the entire deletion or due to model system specific sensitivities. Interestingly, we also found increased ommatidial size with knockdown of *KCTD13*[CG10465], similar to that of *PTEN*[dpten]. This result is consistent with data from Deshpande and colleagues showing increased cell growth in human iPSC-derived neurons from patients with the 16p11.2 deletion[56]. In contrast to increased cell growth in the fly eye, we also found reduced dendritic complexity for *KCTD13*[CG10465], suggesting that the cell growth defects observed with knockdown of individual 16p11.2 genes could also be cell-type specific.

Our results suggest that multiple 16p11.2 homologs contribute to a range of phenotypes that have key roles in different tissue types and organ systems, indicating pleiotropic effects in *Drosophila* that mirror the multitude of phenotypes observed in humans. These data are consistent with other functional studies of 16p11.2 knockdown or knockout models in mouse and zebrafish, which show abnormal neuronal or developmental phenotypes for multiple genes (Supplementary Table 1). Additionally, several 16p11.2 genes have similar intolerance to variation and likelihoods of having loss-of-function mutations compared to causative genes in syndromic CNVs[1], including *RAI1*, *SHANK3*, and *NSD1* (Supplementary Table 3). The presence of multiple genes in 16p11.2 that are individually potentially pathogenic and behave similarly to classical neurodevelopmental genes, as determined by RVIS[57] and pLI scores[58], suggest that interactions between these genes are necessary to modulate the effects of each gene in the deletion. Further, the additive effects of haploinsufficiency of all 16p11.2 genes cannot completely explain the clinical features of the deletion, as all patients with the deletion should then manifest some degree of the affected phenotype. However, this is not the case, as several clinical features of 16p11.2 deletion are not completely penetrant, including autism and macrocephaly[27]. We identified 24 interactions between pairs of 16p11.2 homologs, as well as 64 interactions between the 16p11.2 homologs and key neurodevelopmental genes or genes differentially expressed in one-hit models (Fig. 10). We found that 20 out of 24 interactions between 16p11.2 homologs suppressed the cellular phenotypes observed in one-hit knockdown flies, suggesting that these interactions are epistatic in nature rather than merely additive[43] (Fig. 10, Supplementary Fig. 9a). These results suggest potential complex interactions between the products of 16p11.2 genes at the mechanistic level, which should be confirmed using other model systems.

Based on these results, we propose a pervasive interaction model for the pathogenicity of 16p11.2 deletion, where the phenotypic effects of the whole deletion may not equal the sum of the phenotypic effects due to disruption of its constituent genes. Rather, the interactions between the genes within the deletion, acting through common pathways, determine the phenotypic severity (Fig. 10c). These interactions can suppress, rescue or enhance the one-hit phenotypes, providing evidence for their role towards the incomplete penetrance of clinical features associated with the deletion. The phenotypic variability of the deletion can therefore be explained by variants in upstream regulatory regions or modifier genes that also participate in these pathways (Fig. 10c). In fact, the complex interactions between 16p11.2 genes can amplify the effects of second hits in the genetic background located within the same pathway, as these second hits can potentially modulate the phenotypic effects of several 16p11.2 genes at once (Supplementary Fig. 9b). These second-hits could be targeted in animal or cellular models of the full deletion for reduction of neurodevelopmental phenotypes, which would provide further evidence for the clinical importance of complex interactions among 16p11.2 genes. This model is in contrast to that reported for syndromic CNVs, where the core phenotypes can be due to a single gene (such as *RAI1* in Smith-Magenis syndrome) or a subset of individual genes in the contiguous region (as in Williams syndrome), but agrees with previous findings showing synergistic interactions between genes within de novo CNVs identified in individuals with autism[25]. Our results further suggest the importance of genetic interactions towards causation and modulation of neurodevelopmental disease, and emphasize the need for a function-based analysis in addition to sequencing studies towards discovery of gene function in the context of genetic interactions.

## Methods

**Identification of *Drosophila* homologs.** We first queried the *Drosophila* genome for homologs using DRSC Integrative Ortholog Prediction Tool (DIOPT) (http://www.flyrnai.org/cgi-bin/DRSC_orthologs.pl), reciprocal BLAST and ENSEMBL database for each of the 25 human 16p11.2 genes (Supplementary Table 1). We narrowed down the list of homologs to 15 genes with a DIOPT score of 3 or greater, or in the case of *KIF22*[klp68D], high query coverage and percentage identity in BLAST. Of the 15 selected homologs, RNAi lines were available for all homologs except *INO80E*, and therefore we were unable to characterize this gene. Therefore, 14 homologs of 16p11.2 genes were used in this study. We used a similar strategy for identifying homologs for other genes tested for interactions in this study. We confirmed that all tested 16p11.2 homologs were expressed in the fly eye using database and literature searches (Supplementary Table 1).

***Drosophila* stocks and genetics.** Tissue-specific knockdown of 16p11.2 homologs and other genes tested in this study was achieved with the UAS-GAL4 system, using *w; GMR-GAL4; UAS-Dicer2* (Zhi-Chun Lai, Penn State University), *dCad-GFP, GMR-GAL4* (Claire Thomas, Penn State University), *Elav-GAL4;;UAS-Dicer2* (Scott Selleck, Penn State University), *Elav-GAL4* (Mike Groteweil, VCU), *Da-GAL4* (Scott Selleck, Penn State University), *MS1096-GAL4;; UAS-Dicer2*

---

**Fig. 10** A complex interaction model for pathogenicity of the 16p11.2 deletion. **a** Examples of interactions from quantitative phenotyping data observed with pairwise knockdown of genes. Blue lines indicate modulation of *GeneB* expression in wild-type flies, while orange lines indicate modulation of *GeneA* expression when *GeneB* is also knocked down. *GeneA* knockdowns that have the same phenotype with or without *GeneB* knockdown indicate no interaction between the two genes (left). Epistatic interactions between fly homologs occur when the change in effect for two-hit knockdown flies compared to *GeneA* knockdown is less severe (suppressor) or more severe (enhancer) than that for *GeneB* knockdown compared to control (center). When the effect of *GeneB* knockdown is the same in wild-type flies and flies with *GeneA* knockdown, the two genes show an additive interaction (right). **b** Summary table listing all validated interactions with 16p11.2 *Drosophila* homologs found using screening of eye phenotypes. For epistatic interactions between fly homologs, blue-colored genes represent suppressors while red-colored genes indicate enhancers of the one-hit phenotype. Epistatic interactions with available *Flynotyper* data were confirmed using two-way ANOVA tests ($p < 0.05$, df = 1, $F > 4.5202$; see Supplementary Data 8). Bold genes are annotated for cell proliferation/cell cycle GO terms. *indicates observed cell organization/proliferation defects in the developing eye, and †indicates observed axonal targeting defects. **c** A model of pathogenicity of 16p11.2 deletion inferred from fly studies. The knockdown of individual 16p11.2 homologs in *Drosophila* contributes towards various neuronal or developmental phenotypes. However, pairwise knockdown of 16p11.2 homologs, or knockdown of 16p11.2 homologs with other modifier genes, leads to enhancement, suppression, or rescue of these phenotypes, ultimately resulting in variable phenotypes dependent on the extent of modulation

(Zhi-Chun Lai, Penn State University), *w; C5-GAL4;Dicer2* (Zhi-Chun Lai, Penn State University), and *UAS-RNAi* transgenic lines. The RNAi lines were obtained from the Vienna *Drosophila* Resource Center (VDRC, includes both KK and GD lines)[26] and the Bloomington Stock Center (NIH P40OD018537), and deficiency lines were obtained from the Bloomington Stock Center. All fly stocks and crosses were cultured on conventional cornmeal-sucrose-dextrose-yeast medium at 25 °C unless otherwise indicated. For the eye-specific RNAi knockdown with *GMR-GAL4*, the temperature dependence of *GAL4* activity and knockdown efficiency with *UAS-Dicer2* allowed us to test pathogenicity at varying doses of expression. We set up breeding experiments with *GMR-GAL4* at 30 °C, with or without *Dicer2*, to modulate the level of gene expression. A list of all lines used in this study is presented in Supplementary Data 1.

To study two-locus models, we first generated individual fly stocks with reduced expression for *KCTD13^{CG10465}*, *MAPK3^{rl}*, *PPP4C^{pp4-19C}*, and *DOC2A^{rph}* containing *GMR-GAL4* and *UAS-RNAi* on the same chromosome. We tested to ensure there was adequate *GAL4* to bind to two independent *UAS* constructs in the two-locus models (Supplementary Fig. 1e). Females from the stocks with constitutively reduced gene expression for each of these four genes were then crossed with other RNAi lines to achieve simultaneous knockdown of two genes in the eye. Overall, we performed 565 pairwise knockdown experiments, including 123 interactions between 16p11.2 homologs, 420 interactions with other neurodevelopmental and CNV genes (selected from refs. [35] and [59]), and 22 validation experiments to test interactions with differentially-expressed genes (Supplementary Data 3 and 4).

**RNA extraction and quantitative real-time PCR**. We assessed mRNA expression by performing quantitative real-time PCR (qRT-PCR) experiments on cDNA samples isolated from fly heads. Briefly, RNAi lines were crossed with *Elav-GAL4* driver at 25 °C, and F1 female progeny were collected in groups of 40–50, quickly frozen in liquid nitrogen, and stored at −80 °C. For RNA extraction, the heads were separated from their bodies by repetitive cycles of freezing in liquid nitrogen and vortexing. Total RNA was isolated using TRIZOL (Invitrogen), and reverse transcription was performed using qScript cDNA synthesis kit (Quanta Biosciences). RNA was also isolated from fly heads from *GMR-GAL4* crosses for a subset of genes to compare the gene expression with fly heads from *Elav-GAL4* crosses (Supplementary Fig. 1c). Quantitative RT-PCR was performed using an Applied Biosystems Fast 7500 system with SYBR Green PCR master mix (Quanta Biosciences) using optimized protocols. A list of primers used for the qRT-PCR experiments is provided in Supplementary Data 7.

**Quality control**. We checked the insertion site of the RNAi constructs to identify and remove any fly lines that may show phenotypes due to insertion-site effects (Supplementary Fig. 1a). While RNAi transgenes for the Bloomington lines are inserted at the attP2 site on chromosome 3L with no expression or effect on the nervous system, thorough analysis of RNAi lines obtained from VDRC stock center was required to rule out off-target effects. We obtained two types of lines from VDRC: GD lines, which are P-element-based transgenes with random insertion sites, and KK lines, which are phiC31-based transgenes with defined insertion sites[26]. In order to rule out any effect of insertion of the RNAi construct in the GD lines, we mapped the insertion site by performing Thermal Asymmetric Interlaced PCR (TAIL-PCR) and Sanger sequencing. The TAIL-PCR method was modified from a protocol developed in B. Dickson's lab, based on published protocol[60]. The first round of PCR was performed with a 1:100 dilution of a genomic DNA preparation with Taq polymerase using three degenerate forward primers (AD1, AD2, and AD3) and a specific reverse primer (T1BUAS) (see Supplementary Data 7 for TAIL-PCR primers). The second PCR reaction was set up using 1:50 dilution of the first PCR as template, with the AD primer as the forward primer and T2D as the specific reverse primer. The second PCR products were then visualized in 1% agarose gel, followed by gel extraction of the PCR product, Sanger sequencing using the T2En primer, and analysis of the resulting sequence in BLAST. If the insertion site was in the 5′ UTR, we only excluded the line if there was an overexpression of the downstream gene and the phenotype was discordant with another line. In case of KK lines, Green and colleagues demonstrated that the host strain for the KK library has two landing sites: 5′ UTR of the *tiptop* gene and a previously non-annotated insertion adjacent to 5′ UTR of the *numb* gene (at position chr2L: 9437482, cytological band 30B3)[61]. We observed non-specific shriveled wings in three out of seven KK lines of 16p11.2 homologs with *Elav-GAL4*, and these three lines also showed increased expression of *tiptop* (Supplementary Data 2). Therefore, we excluded these KK lines from neuronal experiments using *Elav-GAL4*. However, we found that overexpression of *tiptop* (using *UAS-tio*) with *GMR-GAL4* showed a rough eye phenotype and reduced pigmentation confined to the right side of the eye, distinct from the eye phenotypes observed in the KK lines (Supplementary Fig. 1d). Further, we did not observe any changes in the expression of *numb* in the fly lines used in this study (Supplementary Data 2).

**Climbing assay**. Fly crosses were set up at 25 °C with *Elav-GAL4* to achieve neuronal knockdown. Four genes, *PPP4C^{pp4-19C}*, *ALDOA^{ald}*, *TAOK2^{dTao}*, and *KIF22^{klp68D}*, showed lethality when neuronal expression was reduced using RNAi at 25 °C, and therefore were tested at room temperature. *KIF22^{klp68D}* lines were also

lethal when raised at room temperature. For each genotype, groups of 10 flies were transferred to a climbing vial and tapped down to the bottom. They were allowed to climb past a line marked at 8 cm from the bottom of the vial, and the number of flies crossing the 8 cm mark at 10 s was recorded as a percentage of flies able to climb per vial (climbing ability). For each group, this assay was repeated nine more times with one-minute rest between each trial. These sets of 10 trials for each group were repeated daily for 10 days, capturing data from flies aged day 1–10. All experiments were performed during the same time of the day for consistency of the results. Two-way ANOVA and pairwise two-tailed *t* tests were used to determine significance for each genotype and day of experiment (Supplementary Data 8).

**Spontaneous seizures assay**. Newly eclosed flies of the relevant genotypes were collected and aged for 7 days. Male and female flies were isolated at least 1 day after collection to ensure all females had mated. After aging, flies were transferred individually into the chambers of a 4 × 5 mating plate using a manual aspirator. The plate was then placed on a standard light box, where the flies were allowed to acclimate for 5 min. Fly behavior was recorded at 30 frames/s for 5 min using a Canon High Definition Vixia HFM31 Camcorder (resolution 1920 × 1080). Each fly's behavior during the viewing window was then assessed for abrupt, involuntary seizure-associated movements, which manifest as rapid repositioning of the flies within the chamber as previously described[28]. The total number of flies that exhibited spontaneous seizure events and the number of seizing events per seizing fly was initially assessed in 10–20 flies with each knockdown genotype (Supplementary Fig. 2b), and validated using 5–7 replicates of 20 flies for three select 16p11.2 homologs (Fig. 2c). Knockdown lines were compared to controls using one-tailed Mann-Whitney tests (number of seizures per fly and percentage of seizing flies in replicate experiments) or Fisher's exact tests (percentage of seizing flies in experiments without replicates) (Supplementary Data 8).

**Dendritic arborization assays**. RNAi lines were crossed to a *UAS-Dicer2; ppk-GAL4, UAS-mCD8-GFP* driver at 25 °C, and embryos were collected at 24 hours on apple juice plates. First instar larvae, eclosed from the embryo, were transferred to the food plate and allowed to age for 48 h at 25 °C before live imaging. Third instar larvae were collected, washed in PBS, and transferred dorsal side up to a glass slide containing a dried agarose pad with a coverslip on top secured with sticky tape. Z-stack images of Class IV Dendritic Arborization neurons were acquired using a Zeiss LSM 800 confocal microscope and processed using ImageJ (https://imagej.nih.gov/ij/) to a scale of 5.0487 pixels/micron. Using an in-house Java plug-in, four concentric circles with a distance of 25 microns between each circle were placed on the images, with the cell body as the center. A manual Sholl analysis was conducted by counting the number of intersections of dendritic branches on each of the concentric circles. Total and average number of intersections were calculated and normalized to the width of the hemisegment of each sampled neuron to control for slight variation in larval sizes. Two-way ANOVA and pairwise two-tailed *t* tests were used to determine significance of the number of intersections in each genotype and concentric circle, and two-tailed Mann-Whitney tests were used to determine significance of the total number of intersections (Supplementary Data 8).

**Phenotypic analysis of fly eyes using *Flynotyper***. We used *GMR-GAL4* drivers with and without *Dicer2* to achieve eye-specific knockdown, and imaged 2–3 day old flies using an Olympus BX53 compound microscope with an LMPlanFL N 20X air objective (Olympus, Tokyo, Japan), at ×0.5 magnification and a z-step size of 12.1 μm. We used CellSens Dimension software (Olympus Optical) to capture the images, and stacked the image slices using Zerene Stacker (Zerene Systems, USA). All eye images presented in the figures are maximum projections of consecutive 20 optical z-sections. Eye area was calculated from each image using ImageJ[62]. Eye phenotypes were scored manually from rank 1–10 based on severity, with rank 1 assigned to wild type-like and rank 10 for the most severe phenotype. We developed a computational method called *Flynotyper* (software available at https://flynotyper.sourceforge.net) that calculates a phenotypic score based on alterations in the hexagonal arrangement of ommatidia in the fly eye[35]. The *Flynotyper* software detects the center of each ommatidium and calculates the phenotypic score based on the number of ommatidia detected, the lengths of six local vectors with direction pointing from each ommatidium to the neighboring ommatidia, and the angle between these six local vectors (Fig. 4a (i)). Using *Flynotyper*, we obtained quantitative measures of fly eye roughness with single gene or pairwise gene knockdown. The significance of *Flynotyper* results compared to a GD control was determined using one-tailed or two-tailed Mann-Whitney tests (Supplementary Data 8). We found no significant differences in *Flynotyper* scores between GD and KK control *Drosophila* lines with and without *Dicer2*, and therefore we used a single control for statistical analysis (Supplementary Fig. 1b). We have previously shown a strong concordance between manual scores and phenotypic scores[35]. In this study, we used manual scoring in conjunction with *Flynotyper*, as certain features such as necrotic patches, glossy eyes, and overall eye size are not detected by *Flynotyper*.

**Immunohistochemistry**. For the neuromuscular synapse (NMJ), female third instar larvae were dissected in 1.8 mM Ca^{2+} and 4 mM Mg^{2+} saline solution

(128 mM NaCl, 2 mM KCl, 1.8 mM $Ca^{2+}$, 4 mM $Mg^{2+}$, 5 mM Hepes, and 36 mM sucrose, pH 7.0) and fixed in saline solution containing 4% paraformaldehyde (PFA) for 30 min. The fixed larvae were washed with saline, PBS (13 mM NaCl, 0.7 mM $Na_2HPO_4$, and 0.3 mM $NaH_2PO_4$), and PBT (0.2% Triton X-100 in PBS) for 10 min each, incubated with blocking buffer (5% normal goat serum in PBT) for one hour, and then incubated with anti-Dlg (1:500; 4F3, Developmental Studies Hybridoma Bank (DSHB), University of Iowa) overnight at 4 °C. These preparations were then washed thrice with PBT and twice with PBS for 6 min each, and incubated with fluorophore-conjugated secondary antibodies, Alexa fluor 568 goat anti-mouse (1:200; A11031, Molecular Probes by Life Technologies), and a plasma membrane marker, Alexa fluor 647-conjugated AffiniPure Goat anti-HRP (1:200; 123-605-021, Jackson ImmunoResearch Laboratories, Inc.), for two hours. Final washes were performed with PBS, five times each for 6 min, and mounted in a 1:1 mixture of PBS and glycerol between two cover slips for imaging.

For the larval and pupal eye disc, the eye discs from wandering third instar or 45-hour-old pupae were dissected in PBS and fixed in PBS containing 4% PFA for 20 min. The tissues were then washed in PBT, treated with blocking solution for 30 min, and then incubated overnight with primary antibodies at 4 °C. Mouse anti-pH3 (S10) antibody (1:200; 9706-Cell Signaling Technology), a specific mitotic marker for measuring proliferating cells, Elav antibody (1:100; 7E8A10, DSHB), a marker for cell differentiation, and mouse anti-chaoptin (1:200; 24B10, DSHB), a marker for retinal axonal projections, were used for larval eye discs, and mouse anti-Dlg (1:200; 4F3, DSHB), a septate junction marker to visualize and count ommatidial cells, and Rhodamine Phalloidin (1:100; R415, Molecular Probes by Life Technologies), an F-actin marker, were used for observing photoreceptor cells in pupal eyes. These preparations were then washed for 10 min thrice with PBT, and incubated with fluorophore-conjugated secondary antibodies (Alexa fluor 568 goat anti-mouse (1:200); A11031; Alexa fluor 488 donkey anti-rat (1:200), A21208; and Alexa fluor 647 goat anti-mouse (1:200); A21236, Molecular Probes by Life Technologies) for two hours. Final washes were performed in PBS, and the tissues were mounted in Prolong Gold antifade reagent with DAPI (Thermo Fisher Scientific, P36930) for imaging.

**BrdU cell proliferation assay.** For BrdU incorporation, the larval eye discs were dissected in PBS and immediately transferred to Schneider media (Sigma). The tissues were then incubated in 10 μM BrdU (Sigma) at 25 °C for 1 h with constant agitation to allow for incorporation of BrdU into the DNA of replicating cells in S phase. The tissues were washed thrice with PBS for 5 min each, and fixed in PBS containing 4% PFA for 20 min. The tissues were acid-treated in 2 N HCl for 20 min to denature DNA. Subsequently, the tissues were neutralized in 100 mM Borax solution for 2 min, washed three times with PBT for 10 min each, and treated with blocking solution for 1 hour. Then, tissues were incubated with mouse anti-BrdU (1:200; DSHB-G3G4) diluted in blocking solution overnight at 4 °C. On the following day, the tissues were washed three times in PBT for 20 min each and incubated in Alexa flour-568 Goat anti-mouse (1:200; A11031) diluted in 1X PBS, containing 5% normal goat serum, for two hours with constant agitation. Finally, tissues were mounted in Prolong Gold antifade reagent with DAPI.

**Confocal microscopy and image analysis.** We acquired Z-stack images of larval eye discs (proliferation assay), pupal eye discs (cellular architecture), and body wall muscles 6 and 7 in the abdominal segments A2 and A3 (NMJ architecture) using an Olympus Fluoview FV1000 laser scanning confocal microscope (Olympus America, Lake Success, NY). Acquisition and processing of images was performed with the Fluoview software (Olympus). We used one or two optical sections for larval and pupal eye disc images, and maximum projections of two or three optical sections were used for NMJ images. For BrdU staining, Elav staining and pro-liferation (anti-pH3) assays, maximum projections of all optical sections were generated for display. Area, length, perimeter, and number of branches in neu-romuscular synapses were calculated using the Drosophila_NMJ_morphometrics macro in ImageJ[63]. The bouton counts in each NMJ and pH3-positive cells from larval tissues were counted using the Cell Counter Plug-In within ImageJ. We also calculated the number of pH3 positive cells using the Analyze Particles function in ImageJ, and found a high correlation (Pearson correlation, $r = 0.9599$, $p < 0.0001$) with counts obtained from Cell Counter Plug-In (Supplementary Fig. 1f). Sig-nificance of cell counts or NMJ features from confocal microscopy compared to GD controls was determined using one-tailed or two-tailed Mann-Whitney tests (Supplementary Data 8).

**Differential expression analysis of transcriptome data.** We performed RNA sequencing of samples isolated from fly heads of $Elav$-$GAL4 > Dicer2$ crosses for $MAPK3^{rl}$, $KCTD13^{CG10465}$, $DOC2A^{rph}$, $CORO1A^{coro}$, $C16ORF53^{pa1}$, and $CDIPT^{pis}$, and compared gene expression levels to VDRC control flies carrying the same genetic background. We prepared cDNA libraries for three biological replicates per knockdown model using TruSeq Stranded mRNA LT Sample Prep Kit (Illumina, San Diego, CA), and performed single-end sequencing using Illumina HiSeq 2000 to obtain 100 bp reads at an average coverage of 35.2 million aligned reads/sample. We used FastQC (www.bioinformatics.babraham.ac.uk/projects/fastqc) and Trimmomatic[64] for quality control assessment, TopHat2[65] v.2.1.0 to align the raw sequencing data to the reference fly genome and transcriptome build 6.08, and

HTSeq-Count[66] v.0.6.1 to calculate raw read counts for each gene. edgeR[67] v.3.16.5 (generalized linear model option) was used to perform differential expression analysis. Genes with a $\log_2$-fold change $>1$ or $<-1$, and with a corrected false-discovery rate less than 0.05, were considered as differentially expressed (Supplementary Data 5). We used the log-fold change in expression to confirm reduced gene expression of each 16p11.2 homolog in the tested RNAi lines. These values were similar to expression values obtained by qPCR; we found a positive corre-lation between qPCR and RNA-Seq derived expression values for 186 differentially expressed genes across the six knockdown models (Pearson correlation, $r = 0.4677$, $p = 1.672 \times 10^{-11}$). Human homologs of differentially-expressed fly genes were identified using DIOPT v.5.3.

**Functional enrichment in differentially expressed genes.** We used gene set enrichment analysis to summarize the genome-wide list of genes and their levels of differential expression into biological pathways and processes perturbed by knockdown of 16p11.2 homologs (Supplementary Fig. 8a). First, we used DIOPT to identify fly homologs of all annotated genes in each human Gene Ontology Biological Process term. We then calculated Z scores for all GO terms with less than 500 genes (in order to exclude very general GO terms) across the six knockdown models using the Parametric Analysis of Geneset Enrichment procedure[48]. This method averages the log-fold change in expression of all genes in every GO term, and then subtracts the mean and divides by the standard deviation of the log-fold change levels in all genes. The Z-score represents the degree of up- or down-regulation of all genes within the GO term. We estimated a p-value for each Z score by comparing to the standard normal distribution (two-sided test), and corrected for multiple hypothesis testing using the Benjamini-Hochberg method. 516 GO terms with corrected p-values < 0.01 are listed in Supplementary Data 5. We also used Cytoscape to visualize the network of cell proliferation (GO:0008283) and cell cycle (GO:0007049) genes that were differentially expressed in the knockdown models (Supplementary Fig. 8b).

**16p11.2 gene interactions in a human brain-specific network.** We used a human brain-specific gene interaction network[50] to further contextualize the observed interactions in 16p11.2 homologs. This network was built using a Bayesian framework that integrated brain-specific signals from genomic data published in over 14,000 publications[50]. Within this network, we mapped 49 genes with identified interactions in the fly eye, and calculated the shortest paths between these genes. This procedure identified 428 additional genes in the network that were critical in connecting the 49 assayed genes to each other (Supplementary Data 6). We then examined these connector genes for enrichment of genes with cell proliferation and cell cycle GO annotations using a one-sided Fisher's exact test.

**Data availability.** Gene expression data for the six 16p11.2 homolog model fly lines that support the findings of this study are deposited in the GEO (Gene Expression Omnibus) database with accession code GSE100387, and the raw RNA-Sequencing files are deposited in the SRA (Sequence Read Archive) with BioProject accession PRJNA391493. All other data supporting the findings of this study are available within the paper and its supplementary information files.

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

## Acknowledgements

We thank T. Mackay, S. Williams, and J. Moore for helpful discussions on the role of epistasis in our fly interaction studies, and E. Eichler, J. Kumar, F. Hormozdiari, D. Cavener, M. Grotewiel, C. Shashikant, Z-C. Lai, V. Hoxha, and S. Selleck for useful discussions and critical reading of the manuscript. We also thank R. Danjo and R. Pandya for assistance with fly husbandry, I. Albert, A. Sebastian, and Q. Li for assistance with RNA-Sequencing data analysis, A. Castells and A. Schenck for help with NMJ morphometrics, and

A. Srivastava for technical help. This work was supported by a Basil O'Connor Award from the March of Dimes Foundation (#5-FY14-66), NIH R01-GM121907, a NARSAD Young Investigator Grant from the Brain and Behavior Research Foundation (22535), and resources from the Huck Institutes of the Life Sciences to S.G., and NIH T32-GM102057 to M.J.

## Author contributions

J.I., M.D.S., M.J., and S.G. designed the study. J.I., M.D.S., P.P., L.P., E.H., P.L., K.V., A.K., Q.W., A.T., S.Y., and J.B. performed the experiments. A.T.W. and M.M.R. performed dendritic arborization experiments, and H.K. and J.R.M. performed epilepsy phenotyping. M.J. and A.K. performed network analysis. J.I., M.D.S., M.J., and S.G. analyzed data. J.I., M.J., and S.G. wrote the manuscript with input from all authors.

## Additional information

**Competing interests:** The authors declare no competing interests.

