## [Peer Review File · Nature Communications]

Reviewers' comments:

Reviewer #1 (Remarks to the Author):

I am a previous reviewer of this manuscript.

I commend the authors on the bevy of additional data and their thorough consideration of the comments. My questions have mostly been answered, to the point of being largely satisfied with this version of the manuscript. I do, however, feel the need to make one comment.

Other reviewers had very substantive concerns that were argued away in the review. However, I think the authors should really think about putting some of that justification in the manuscript. This paper has always suffered from incomplete or unclear explanation. The more logic behind why the experiments were done and why the data were interpreted as they were that goes into the final paper, the better.

Reviewer #2 (Remarks to the Author):

I would like to thank Giriajan and colleagues for the efforts they have put into addressing my concerns. I am also extremely grateful to them for the clear way they have laid out their arguments and revisions in the document for reviewers. Thank you, it is much appreciated (Editor note this!!)

I believe the revisions they have made have significantly enhanced their paper (quite an opus) and provided clarity on several issues. There is much to commend here and an awful lot of ideas explored that will be of great value to the 16p11.2 community and beyond. I commend them. My only hope is that the many excellent analyses and approaches they have reported have not been buried, but rather that a researcher drawn to their paper for one thing, might read the many approaches and leave with lots of new ideas.

I have, I'm afraid, one remaining problem. This is not the paper in which the meaning of the term "epistasis" should be changed. This is important. Much of the research around epistasis is centred around modelling independently segregating loci and if the meaning of the word is changed then this will create huge confusion and inhibit community cross-talk. I sympathise with the authors in wishing to harness the interest in epistasis to draw readers in but it is wrong and it is misleading.

If the authors are so keen to have “epistasis” in the title then why not “Epistasis amongst 16p11.2 fly orthologs demonstrate genetic interactions modulate disease” or something similar?

If the authors are serious about a need to change the meaning of the word epistasis then I suggest they write an opinion piece laying out the need for this change in order to progress our understanding of human disease. Personally, I think “genetic interactions” provides sufficient clarity.

I again repeat my request that the authors keep the usage of epistasis to interaction between distinct fly loci and use genetic interaction when addressing the human 16p11.2 genes. I think this is important.

Reviewer #3 (Remarks to the Author):

The authors have adequately addressed my previous comments. My remaining concerns are described below.

1. The current study has looked extensively at gene main effects and pairwise interactions on a wide variety of anatomical and neurodevelopmental phenotypes in drosophila. The results reported here are a beautiful first step in beginning to elucidate how the effects of dozens of genes may influence clinical phenotype in humans. However, the relationship of specific phenotypes and effects to the human clinical phenotype or phenotypes of the deletion in mouse are not clear. Mouse models would be beyond the scope of this study, but further studies of models of the full deletion will still be needed to bridge from drosophila to human. As such, I think the title “Pervasive epistasis modulates neurodevelopmental defects of the autism-associated 16p11.2 deletion” (and other similar claims) are overstated and should be rephrased

The full deletion was not modeled in drosophila and specific gene effects detected in this study were not confirmed in a full deletion model in another organism. Furthermore this study does not quantify how much of the full deletion effect represents additive vs epistatic effects. A more appropriate title would be something like “Genetic screens in drosophila identify neurodevelopmental phenotypes and epistatic effects among genes within the 16p11.2 deletion”

2. This paper describes a large amount of work, and it’s challenging for a reader to digest all of it. Figures are very dense. Results mentioned in a single sentence refer to both main and supplemental figures and references to specific figures/panels are often placed at the beginning or end of a sentence making it unclear which result corresponds to which figure/panel. The manuscript needs to be edited for clarity so that the reader can easily find each result that’s mentioned in the text. Here are just a few examples. But many (in fact most) of the results are presented in this way which is confusing.

Line 127 “We identified dramatic reductions in climbing ability of ALDOA and MAPK3...”. Since the results are in 2 different figures, cite each one separately: ALDOA (Fig S2A) and Mapk3 (Fig 2B). Red data label is missing from the legend of Fig S2A.

Line 184 The following results are spread across Figures 3E, 3F and S4 but no specific references are provided “KCTD13 CG10465 knockdown flies showed a drastic increase in the number of cone cells, secondary pigment cells, and photoreceptor neurons, while MAPK3rl knockdown showed a decreased number of interommatidial cells and photoreceptor neurons, with a consequent loss of the hexagonal structure in the ommatidia. Similarly, ALDOAald knockdown flies had misplaced bristle cells as well as an increase in secondary and photoreceptor cells, while PPP4Cpp4-19C knockdowns showed severe rotational defects and a complete loss of the ommatidial architecture.”

3. Fig 2C legend please clarify 5-7 replicate samples containing 20 flies each –or- 20 samples containing 5-7 replicates?

4. Two-locus fly models (line 227). It’s unclear what exact (statistical?) criteria is being used for inferring whether a genetic interaction is in fact occurring. It does not appear that interaction effects are being tested in their ANOVA model. When the main effect of a gene is significantly greater or less when combined with a 2nd gene that has no clear effect of its own, then the interaction test may be significant, but this needs to be formally tested.

It seems that main argument for epistasis is the observation that some genes SUPPRESS the phenotype of another gene. This is reasonable, but it should still be tested statistically.

Fig 7. Notably, most of the interactions reported consisted of suppression and there were very few additive or enhancing effects, but this result is likely biased due the criteria that were used. A more unbiased statistical approach would likely find a more even number of enhancing and suppressing effects and quite of the pairwise gene effects may actually be consistent with additivity.

5. Line 285: “Interactions of 16p11.2 homologs within cell proliferation pathways determine the pathogenicity of the phenotype”. Since it’s unclear how most of the specific phenotypes in this paper relate to the human clinical phenotype, it’s misleading to use the word “pathogenicity”. “Severity” of the phenotype would be more appropriate.

6. Identification of genes (e.g. CHD8, PTEN) that suppress phenotypes of 16p11.2 genes highlights potential therapeutic mechanisms. Further studies in higher organisms are needed to determine whether modulation of these genes could rescue phenotypes of the full deletion. It’s overstating things a bit to refer to these as promising therapeutic targets in the abstract and discussion. I recommend to make the statements more nuanced.

7. As mentioned above, follow up studies of these effects need to be done in full deletion models. What is needed to really demonstrate clinical relevance? This should be addressed in the discussion.

REVIEWERS' COMMENTS:

Reviewer #1:

- 1. Other reviewers had very substantive concerns that were argued away in the review. However, I think the authors should really think about putting some of that justification in the manuscript. This paper has always suffered from incomplete or unclear explanation. The more logic behind why the experiments were done and why the data were interpreted as they were that goes into the final paper, the better.*

Response: We thank the reviewer for this suggestion on improving the manuscript. We have carefully reviewed previous and current reviews and have adequately addressed these concerns in the revised manuscript. Compared to our original submission, we have expanded the Discussion and added multiple figures to the manuscript to improve the presentation of our data and also to clarify specific details. For example, we have provided additional clarifications on the statistical analysis of interaction data, generated a summary of all identified interactions (Figure 10b), and provided an additional schematic to illustrate our interpretation of the results (Supplementary Figure 9b).

Reviewer #2:

This is not the paper in which the meaning of the term “epistasis” should be changed. This is important. Much of the research around epistasis is centred around modelling independently segregating loci and if the meaning of the word is changed then this will create huge confusion and inhibit community cross-talk. I sympathise with the authors in wishing to harness the interest in epistasis to draw readers in but it is wrong and it is misleading.

1. *If the authors are so keen to have “epistasis” in the title then why not “Epistasis amongst 16p11.2 fly orthologs demonstrate genetic interactions modulate disease” or something similar?*

Response: We acknowledge the concerns of the reviewer in using the term “epistasis” in the title of our manuscript. We have decided to modify our title to instead read “*Pervasive genetic interactions modulate neurodevelopmental defects of the autism-associated 16p11.2 deletion*”.

2. *If the authors are serious about a need to change the meaning of the word epistasis then I suggest they write an opinion piece laying out the need for this change in order to progress our understanding of human disease. Personally, I think “genetic interactions” provides sufficient clarity. I again repeat my request that the authors keep the usage of epistasis to interaction between distinct fly loci and use genetic interaction when addressing the human 16p11.2 genes.*

Response: We agree with the reviewer that “complex genetic interactions” can be used to avoid any confusion that could arise from using “epistasis” in the context of the human deletion. To address this, we have removed all references to “epistasis” when discussing the human 16p11.2 deletion in the Discussion, including removing one sentence on how the term could be applied to the 16p11.2 deletion. We have replaced the term “epistasis” with “genetic interactions” or “complex interactions”:

Discussion, page 17: “These results suggest potential complex interactions between the products of 16p11.2 genes at the mechanistic level, which should be confirmed using other model systems. ... These interactions can suppress, rescue or enhance the one-hit phenotypes, providing evidence for their role towards the incomplete penetrance of clinical features associated with the deletion.”

Following the reviewer’s suggestion, we have kept the term “epistasis” to describe non-additive pairwise interactions in the context of the *Drosophila* 16p11.2 homologs. For example, we have modified the legend of Figure 10 to emphasize that we only describe epistasis between pairs of *Drosophila* homologs (changes underlined):

Figure 10 legend, pages 40-41: “Epistatic interactions between fly homologs occur when the change in effect for two-hit knockdown flies compared to *GeneA* knockdown is less severe (suppressor) or more severe (enhancer) than that for *GeneB* knockdown compared to control (center). ... Summary table listing all validated interactions with 16p11.2 *Drosophila* homologs found using screening of eye phenotypes. For epistatic interactions between fly homologs, blue-colored genes represent suppressors while red-colored genes indicate enhancers of the one-hit phenotype.”

Reviewer #3:

1. *The results reported here are a beautiful first step in beginning to elucidate how the effects of dozens of genes may influence clinical phenotype in humans. However, the relationship of specific phenotypes and effects to the human clinical phenotype or phenotypes of the deletion in mouse are not clear. Mouse models would be beyond the scope of this study, but further studies of models of the full deletion will still be needed to bridge from Drosophila to human. As such, I think the title “Pervasive epistasis modulates neurodevelopmental defects of the autism-associated 16p11.2 deletion” (and other similar claims) are overstated and should be rephrased. The full deletion was not modeled in Drosophila and specific gene effects detected in this study were not confirmed in a full deletion model in another organism. Furthermore this study does not quantify how much of the full deletion effect represents additive vs. epistatic effects. A more appropriate title would be something like “Genetic screens in Drosophila identify neurodevelopmental phenotypes and epistatic effects among genes within the 16p11.2 deletion”*

Response: We acknowledge the reviewer’s concern in using “epistasis” in the title of our manuscript and in applying the term in the context of the human deletion. We also agree that these interactions should be tested in a deletion model to fully ascertain the effects of these complex interactions on the variable deletion phenotypes. To alleviate these concerns, we have modified our title to read “*Pervasive genetic interactions modulate neurodevelopmental defects of the autism-associated 16p11.2 deletion*”. We have also replaced all references to “epistasis” with “complex genetic interactions” in the Discussion, and have added that these interactions should be tested with other model systems before they can be definitively associated with the human deletion:

Discussion, page 17: “These results suggest potential complex interactions between the products of 16p11.2 genes at the mechanistic level, which should be confirmed using other model systems.”

2. *This paper describes a large amount of work, and it’s challenging for a reader to digest all of it. Figures are very dense. Results mentioned in a single sentence refer to both main and supplemental figures and references to specific figures/panels are often placed at the beginning or end of a sentence making it unclear which result corresponds to which figure/panel. The manuscript needs to be edited for clarity so that the reader can easily find each result that’s mentioned in the text. Here are just a few examples. But many (in fact most) of the results are presented in this way which is confusing.*

Line 127: “We identified dramatic reductions in climbing ability of ALDOA and MAPK3...”. Since the results are in 2 different figures, cite each one separately: ALDOA (Fig S2A) and Mapk3 (Fig 2B). Red data label is missing from the legend of Fig S2A.

Line 184: The following results are spread across Figures 3E, 3F and S4 but no specific references are provided: “KCTD13^{CG10465} knockdown flies showed a drastic increase in the number of cone cells, secondary pigment cells, and photoreceptor neurons, while MAPK3^{rl} knockdown showed a decreased number of interommatidial cells and photoreceptor neurons, with a consequent loss of the hexagonal structure in the ommatidia. Similarly, ALDOA^{ald}

knockdown flies had misplaced bristle cells as well as an increase in secondary and photoreceptor cells, while PPP4C^{pp4-19C} knockdowns showed severe rotational defects and a complete loss of the ommatidial architecture.”

Response: We apologize to the reviewer that our figure citations in the Results lacked clarity. We have reviewed the Results section and have added or moved citations when needed to assign each result to its specific figure, especially when describing *Flynotyper* and cellular phenotyping data for two-hit interaction studies. Following are three examples of sentences with modified citations, including the two mentioned by the reviewer:

Results, page 5: “We performed negative geotaxis experiments to measure locomotor function and identified dramatic reductions in the climbing ability of *MAPK3^{rl}* (Figure 2b) and *ALDOA^{ald}* (Supplementary Fig. 2a) knockdown flies throughout the testing period.”

Results, page 7: “For example, *KCTD13^{CG10465}* knockdown flies showed a drastic increase in the number of cone and secondary pigment cells (Figure 5b) and photoreceptor neurons (Figure 5d), while *MAPK3^{rl}* knockdown showed a decreased number of photoreceptor neurons (Figure 5d) and interommatidial cells, with a consequent loss of the hexagonal structure in the ommatidia (Figure 5b). Similarly, *ALDOA^{ald}* knockdown flies had misplaced bristle cells (Figure 5b) as well as an increase in secondary pigment cells and photoreceptor neurons (Figure 5d), while *PPP4C^{pp4-19C}* knockdowns showed severe rotational defects and a complete loss of the ommatidial architecture (Figure 5b).”

Results, page 12: “We observed a complete rescue of defects in cellular organization (Supplementary Fig. 7f), photoreceptor cell counts (Figure 7e) and cell proliferation (Figure 7f) observed with *MAPK3^{rl}* single-hit knockdown.”

We also thank the reviewer for observing the oversight of the missing data label in Supplementary Figure 2a (negative geotaxis for *ALDOA^{ald}*); we have corrected this in the revised manuscript.

3. *Fig 2C legend: please clarify 5-7 replicate samples containing 20 flies each –or- 20 samples containing 5-7 replicates?*

Response: For the seizure assay described in Figure 2c, we used between 5-7 replicate samples of 20 flies each. We have clarified this in the revised figure legend:

Figure 2 legend, page 36: “Assessment of knockdown of 16p11.2 homologs for frequency of spontaneous unprovoked seizure events (n=5-7 replicate groups of 20 flies each) and average number of seizure events per fly (n=52-101 individual flies, Mann-Whitney test, *p<0.05).”

4. *Two-locus fly models (line 227). It’s unclear what exact (statistical?) criteria is being used for inferring whether a genetic interaction is in fact occurring. It does not appear that interaction effects are being tested in their ANOVA model. When the main effect of a gene is significantly*

greater or less when combined with a 2nd gene that has no clear effect of its own, then the interaction test may be significant, but this needs to be formally tested.

It seems that main argument for epistasis is the observation that some genes SUPPRESS the phenotype of another gene. This is reasonable, but it should still be tested statistically.

Fig 7: Notably, most of the interactions reported consisted of suppression and there were very few additive or enhancing effects, but this result is likely biased due the criteria that were used. A more unbiased statistical approach would likely find a more even number of enhancing and suppressing effects and quite of the pairwise gene effects may actually be consistent with additivity.

Response: We thank the reviewer for seeking clarification for how we determined genetic interactions. We consulted with Dr. Trudy Mackay (an expert researcher on epistatic models) to infer epistasis and additive interactions based on *Flynotyper* scores. Here, based on the reviewer's suggestion, we performed two-way ANOVA tests for all interactions with available *Flynotyper* data (65 total interactions) and found significant differences for all suppressors and most enhancers, indicative of epistatic interactions. However, we found that ANOVA tests for *KCTD13*^{CG10465}/*PPP4C*^{pp4-19C} and *KCTD13*^{CG10465}/*MAPK3*^{rl} interactions were not significant, and therefore are additive instead of epistatic. We have revised Figure 10b (previous Figure 7b; shown below as Figure R1) to reflect this change. The details of the two-way ANOVA tests are presented in Supplementary Data 8 (statistics analysis). We have also revised Supplementary Fig. 9a (previous Supplementary Fig. 8a; shown below as Figure R2) in order to show *Flynotyper* plots of suppressors as well as enhancers. Overall, these statistical tests support our model of numerous additive and epistatic interactions among 16p11.2 homologs in *Drosophila*.

	Total interactions		Epistatic interactions			Additive interactions	
	Tested	Validated	16p11.2 genes	Neurodevelopmental genes	RNA-Seq targets	16p11.2 genes	Neurodevelopmental genes
KCTD13 ^{CG10465}	45	23	6: C16ORF53 ^{pat} , CDIPT ^{tbls} , FAM57B ^{CG17841} , CORO1A ^{coro1} , ALDOA ^{ald} *, YPEL3 ^{CG15309}	5: CCDC101 ^{19F29} , UBE3A ^{tblbx3A} , CHD8 ^{tbls} , SCN1A ^{para} , SHANK3 ^{proasp}	8: CNGA2 ^{CG42260} , RAF1 ^{CG14607} †, CYP24A1 ^{19p12st-d} , IGFALS ^{CG18035} , GNS ^{CG30059} , ZNF160 ^{pp84ag} , PLEC ^{tbls} , METTL6 ^{CG34195}	1: PPP4C ^{pp4-19C}	3: ATXN2L ^{tbls2} , CHRNA7 ^{tbls} , CTNNB1 ^{tbls}
MAPK3 ^{rl}	47	39	7: C16ORF53 ^{pat} , CDIPT ^{tbls} , FAM57B ^{CG17841} , CORO1A ^{coro1} , ALDOA ^{ald} *, YPEL3 ^{CG15309} , TBX6 ^{tbls3}	18: CCDC101 ^{19F29} , TUFA1 ^{tbltbls} , SH2B1 ^{tbls} , SPNS1 ^{tbls} , PTEN ^{tbls} *, CHD8 ^{tbls} , SCN1A ^{para} , SHANK3 ^{proasp} , EPHA6 ^{tbls} , LGR5 ^{tbls} , NRXN1 ^{tbls} , CEP135 ^{tbls152} , CENPJ ^{tbls4} , TUBGCP6 ^{tbls128} , ASPM ^{tbls} , NIPA2 ^{tbls} , CHRNA7 ^{tbls} , CTNNB1 ^{tbls}	10: COX6A2 ^{tbls4L} †, PIH1D3 ^{CG05048} , FRRS1 ^{CG14513} , LIPA ^{CG08753} , RTN4RL2 ^{tbls} , CECR1 ^{tbls42} , ADCY7 ^{CG32091} , ABCBC5 ^{CG43672} , HOXD4 ^{tbls} , PGCb ^{tbls}	1: PPP4C ^{pp4-19C}	3: KIF11 ^{tbls6F} , LRRC33 ^{CG07895} , BDH1 ^{CG8888}
PPP4C ^{pp4-19C}	37	13	5: C16ORF53 ^{pat} , CDIPT ^{tbls} , CORO1A ^{coro1} , DOC2A ^{tbls} , YPEL3 ^{CG15309}	2: SHANK3 ^{proasp} , CHRNA7 ^{tbls}	--	2: KCTD13 ^{CG10465} , MAPK3 ^{rl}	4: ATXN2L ^{tbls2} , ATP2A1 ^{tbls} , PTEN ^{tbls} , CTNNB1 ^{tbls}
DOC2A ^{tbls}	37	13	2: CDIPT ^{tbls} , ALDOA ^{ald}	9: CCDC101 ^{19F29} , TUFA1 ^{tbltbls} , SPNS1 ^{tbls} , SCN1A ^{para} , SHANK3 ^{proasp} , LGR5 ^{tbls} , NRXN1 ^{tbls} , NIPA2 ^{tbls} , BCL9 ^{tbls}	--	0	2: ATXN2L ^{tbls2} , CTNNB1 ^{tbls}

Figure R1. Summary table listing all validated interactions with 16p11.2 *Drosophila* homologs found using screening of eye phenotypes. For epistatic interactions between fly homologs, blue-colored genes represent suppressors while red-colored genes indicate enhancers of the one-hit phenotype. Epistatic interactions with available *Flynotyper* data were confirmed using two-way ANOVA tests (p<0.05, df=1, F>4.5202). Bold genes are annotated for cell proliferation/cell cycle GO terms. * indicates observed cell organization/proliferation defects in the developing eye, and † indicates observed axonal targeting defects.

Figure R2. Representative *Flyotyper* phenotypes of 16p11.2 pairwise knockdowns showing enhancement or suppression of the one-hit phenotypes in the *Drosophila* eye. Error bars indicate mean \pm standard deviation of *Flyotyper* scores for the selected genotypes (n=6-14 samples). Epistatic enhancer and suppressor interactions were confirmed using two-way ANOVA tests ($p < 0.05$, $df=1$, $F > 4.5202$).

- Line 285: “Interactions of 16p11.2 homologs within cell proliferation pathways determine the pathogenicity of the phenotype”. Since it’s unclear how most of the specific phenotypes in this paper relate to the human clinical phenotype, it’s misleading to use the word “pathogenicity”. “Severity” of the phenotype would be more appropriate.

Response: We agree with the reviewer that this section heading may have been confusing to the reader. Due to *Nature Communications* formatting guidelines (<60 characters for section headings), we have changed this heading to read “16p11.2 homologs interact with known neurodevelopment genes.”

- Identification of genes (e.g. *CHD8*, *PTEN*) that suppress phenotypes of 16p11.2 genes highlights potential therapeutic mechanisms. Further studies in higher organisms are needed to determine whether modulation of these genes could rescue phenotypes of the full deletion. It’s overstating things a bit to refer to these as promising therapeutic targets in the abstract and discussion. I recommend to make the statements more nuanced.

Response: We agree with the reviewer that further studies on drugs targeting interacting genes should be performed on model systems for the full deletion. We have removed the mention of therapeutic drug targets from the abstract and have revised the related Discussion section (changes underlined):

Abstract, page 2: “Our study indicates a role for pervasive genetic interactions within CNVs towards cellular and developmental phenotypes.”

Discussion, page 15: “Screening for interactions with neurodevelopmental genes and differentially-expressed genes in the transcriptome could be particularly useful in identifying potential therapeutic targets for 16p11.2 deletion phenotypes. ... Therefore, therapeutic targets for the identified suppressors of multiple 16p11.2 homologs both within and outside the region, such as ALDOA, CORO1A, CHD8, PTEN, and RAF1, could be tested in full deletion models for a reduction in severity of neurodevelopmental phenotypes. This approach would be especially well suited for 16p11.2 deletion, where genes participating in a shared pathway can be targeted by a single treatment (instead of multiple targets for individual CNV genes).”

7. *As mentioned above, follow up studies of these effects need to be done in full deletion models. What is needed to really demonstrate clinical relevance? This should be addressed in the discussion.*

Response: We agree with the reviewer that further studies of the full deletion model would be necessary to confirm that complex interactions between 16p11.2 genes have clinical relevance to the human deletion. One demonstration of clinical relevance would be to target genes participating in a shared pathway with 16p11.2 genes, such as cell proliferation, and observe any resulting reduction in neurodevelopmental phenotype severity. For example, Deshpande *et al.* (*Cell Rep.* 2017) recently described cell growth defects in iPSC-derived neurons from patients with 16p11.2 deletion, which could be rescued by targeting a cell proliferation gene such as *PTEN* with therapeutic or gene editing approaches. At an organism level, genes in shared pathways could also be targeted in mouse deletion models to detect changes in the observed behavioral phenotypes, such as hyperactivity or motor defects (Portmann *et al.*, *Cell Rep.* 2014). These studies would provide evidence for the importance of interactions among 16p11.2 genes in conserved pathways towards pathogenicity and potential treatment of clinical phenotypes. We have summarized this point in the Discussion of the manuscript:

Discussion, page 17: “These second-hits could be targeted in animal or cellular models of the full deletion for reduction of neurodevelopmental phenotypes, which would provide further evidence for the clinical importance of complex interactions among 16p11.2 genes.”